# Impedance Iterative Learning Backstepping Control for Output-Constrained Multisection Continuum Arms Based on PMA

**DOI:** 10.3390/mi13091532

**Published:** 2022-09-16

**Authors:** Yuexuan Xu, Xin Guo, Jian Li, Xingyu Huo, Hao Sun, Gaowei Zhang, Qianqian Xing, Minghe Liu, Tianyi Ma, Qingsong Ding

**Affiliations:** 1School of Artificial Intelligence and Data Science, Hebei University of Technology, Tianjin 300400, China; 2Engineering Research Center of Intelligent Rehabilitation Device and Detection Technology, Ministry of Education, School of Artificial Intelligence and Data Science, Hebei University of Technology, Tianjin 300400, China; 3School of Automation, Beijing University of Posts and Telecommunications, Beijing 100876, China; 4Beijing Key Laboratory of Rehabilitation Technical Aids for Old-Age Disability and Key Laboratory of Neuro-Functional Information and Rehabilitation Engineering of the Ministry of Civil Affairs, National Research Center for Rehabilitation Technical Aids, Beijing 100176, China; 5Department of Cardiology (Ward 1), Tianjin Chest Hospital, Tianjin 300202, China; 6School of Electronic Information and Automation, Civil Aviation University of China, Tianjin 300300, China

**Keywords:** multisection continuum arms, constant curvature model, adaptive ILC with initial error, barrier Lyapunov function, ANSYS/ADAMS/MATLAB

## Abstract

Background: Pneumatic muscle actuator (PMA) actuated multisection continuum arms are widely applied in various fields with high flexibility and bionic properties. Nonetheless, their kinematic modeling and control strategy proves to be extremely challenging tasks. Methods: The relationship expression between the deformation parameters and the length of PMA with the geometric method is obtained under the assumption of piecewise constant curvature. Then, the kinematic model is established based on the improved D-H method. Considering the limitation of PMA telescopic length, an impedance iterative learning backstepping control strategy is investigated. For one thing, the impedance control is utilized to ensure that the ideal static balance force is maintained constant in the Cartesian space. For another, the iterative learning backstepping control is applied to guarantee that the desired trajectory of each PMA can be accurately tracked with the output-constrained requirement. Moreover, iterative learning control (ILC) is implemented to dynamically estimate the unknown model parameters and the precondition of zero initial error in ILC is released by the trajectory reconstruction. To further ensure the constraint requirement of the PMA tracking error, a log-type barrier Lyapunov function is employed in the backstepping control, whose convergence is demonstrated by the composite energy function. Results: The tracking error of PMA converges to 0.004 m and does not exceed the time-varying constraint function through cosimulation. Conclusion: From the cosimulation results, the superiority and validity of the proposed theory are verified.

## 1. Introduction

Bionics is inspiring researchers in a growing number of fields, and bionic knowledge is applied by continuum robotics more clearly and intuitively than in other areas of research. Due to the limbs of various mollusks exhibiting extraordinary locomotion, manipulation, and dexterity in complex and restricted environments, more and more flexible materials are utilized to mimic the locomotion mechanisms of mollusk organs, such as an elephant’s trunk and octopus’ arms in nature, to create new types of bionic robots, also known as continuum robots, for various applications [1]. A continuum robot can be defined as a robot consisting of a rubber structure that is capable of continuous bending with infinite degrees of freedom. By contrast to conventional robotic arms, continuum robots do not possess discrete joints [2], such that they have the competence to change their original shape and size by altering the elongation and bending joints to adapt to unstructured environments. Continuum robots can be applied in agriculture to pick fruits and vegetables [3]; in industry to carry objects [4]; and in medicine as neuroendoscopic [5] and colonoscopic robots [6], which have promising application prospects. Through the study of the drive approach of continuum robots, it is found that the pneumatic muscle is a lightweight actuator. Compared with other actuators, its significant advantages are mainly reflected in the excellent bionic performance, flexibility, simple structure, high compressibility ratio, and low production price of pneumatic muscle [7]. From the perspective of future development, PMA will become an extremely pivotal part of flexible actuators, which is especially suitable for multisection continuum robots.

Before constructing a reasonable control algorithm, a kinematic model of a continuum robot needs to be established, which is necessarily more complicated and challenging than a conventional robot with a small number of rigid joints. Some valuable methods have been proposed by researchers in kinematic modeling of continuum robots, such as the Cosserat beam theory [8,9,10], which can provide a more accurate description of the deformation of a continuum robot by taking material properties, load, and gravity into account. The curvilinear equation of motion for an octopus tentacle was designed by Ivanescu et al. when coiling around an object by analyzing the coiling action of an octopus [11]. An accurate geometric model was developed by Rucker et al. to simulate the motion of a concentric tube continuum of a robot arm [12]. Anderson equated the continuum robot with a multisection curve in order to approximate the deformation of each section of the continuum robot [13]. The solution of higher-order partial differential equations and the complicated knowledge of material mechanics were involved in most of the above models. For multisection continuum robots, the real-time control and calculation amount will be dramatically affected. Single hidden-layer linear regression networks were designed by Reinhart and Jochen to learn the kinematics of continuum robots [14]. Nonetheless, this class of kinematic models obtained through autonomous learning cannot describe the specific deformations of continuum robots. Due to the structural characteristics of the continuum robot, the deformation of the continuum constitutive section is close to the constant curvature deformation in the case of no load and small load. The assumption based on constant curvature deformation has become the mainstream approach for kinematic modeling of continuum robots, which not only greatly simplifies the difficulty of describing the deformation of continuum constitutive joints, but also facilitates the calculation of robot poses in real time [15,16,17,18]. Among them, substantial fundamental research has been conducted by Walker’s team to lay down the theoretical framework of the constant curvature assumption [19]. Novel modal kinematics are presented in the literature [20], stating that the numerical singularity is bypassed in the constant curvature assumption, and the model can be extended to the multisection continuum robots with recursive methods.

Aiming at enhancing the dynamic motion of robots, a growing number of continuum robot dynamics models have been proposed by researchers, which give the general spatial dynamics form of multisection continuum robots in the form of Euler–Lagrange equations. For instance, an approach for modeling multisection continuum robot dynamics is presented by Webster et al. based on the center-of-gravity method [21]. A common drawback of this method is that it does not sufficiently consider the dynamic changes induced by the environment on the robot [22,23]. Furthermore, the computational complexity of the dynamics model grows exponentially with the increase in the number of driven joints. Therefore, it is quite difficult to construct accurate and general dynamic models for different structures of continuum robots.

For tracking control problems under unknown model parameters, iterative learning control (ILC) has proven to be an extremely practical and efficient model-free control strategy, especially for periodic and repetitive tasks [24]. The basic principle of iterative learning is to utilize the input-output information of a previous system to revise the system parameters, which in turn generates new control inputs. A proposed iterative learning backstepping controller is constructed by [25] combining it with a second-order sliding mode control theory, which is capable of tracking the desired trajectory with certain constraints for a class of nonlinear MIMO systems. Nevertheless, during the design of iterative learning controllers, specific conditions usually need to be satisfied.

(1)The initial tracking error of the system required is zero.(2)Expected output is a priori.(3)The experiment must be completed within a limited timeframe.

Condition (1) is considered as an assumption by most of the papers. However, the initial tracking error inevitably exists in applications limited by the reset accuracy, such that it is essential to study ILC with an initial state error. An ILC based on the initial signal correction method is employed by [26] to relax the initial value condition of the ILC. Unfortunately, a large tracking error is generated during the signal correction. Since the expansion and contraction of the pneumatic muscle are constrained, the tracking error must be limited to a given range to guarantee that the pneumatic muscle moves under the working conditions [27]. Therefore, it is indispensable to discuss ILC under output-constrained conditions for a PMA-type continuum robot. An output-constrained control for nonlinear systems is presented in [28] by designing an asymmetric time-varying barrier Lyapunov function (BLF) to ensure the constraints. The power-added integral control is combined with a novel segmental BLF [29], which is applied to a high-order Hessenberg-type nonlinear system containing output constraints in order to effectively solve the asymptotic quiescence problem of the aforementioned system.

Another essential problem in addressing the tracking error convergence in a nonlinear system is the permanent unmodeled uncertainties and nonparametric disturbances. As an accepted constraint solution, an integral-type BLF is deployed to fulfill the output constraint in [30]. As to enhance robustness, unknown nonlinear functions and lumped disturbances are handled by a radial basis function neural network (RBFNN) control and robust control. The derivative of the robust control gain is estimated by a finite-time differentiator. [31] investigated a BFASM controller under the condition of actuator saturation and uncertainties. That is to say, compared to a conventional sliding mode control, the upper bound information of disturbance is not required in the aforementioned method. However, this approach does not deal well with the chatter phenomenon near the equilibrium point. In the context of a multisection continuum robot, based on PMA, a few research questions have been addressed with the BLF method from the authors’ research findings. Nevertheless, as to ensure the safe operation of PMA in the convergence process, especially in the ILC strategy with insufficient iterations, BLFs have the competence to be implemented to guarantee the tracking errors are within an acceptable boundary.

Inspired by the above work, a complete set of kinematic modeling is proposed for a class of PMA-based multisection continuum robots. Meanwhile, an impedance iterative learning backstepping control strategy with unknown parameters of the dynamics model is implemented. The ILC is adopted to estimate the unknown parameters of the multisection continuum arms, while an adaptive algorithm-based approach is responsible for handling external disturbances and interaction torques. What is more is that a BLF is implemented to ensure that the tracking error is bounded throughout the iterative process, considering the time-varying constraint on the tracking error. The aforementioned theories are especially adopted for nonrepetitive tasks in which the operative range of the actuator is required to be within a certain limit. The proposed theories can be extended to various application scenarios, such as rehabilitation training, industrial robots responsible for handling and grinding, exploration robots, and agricultural picking and irrigation robots. Hence, the main contributions of this paper can be summarized as follows:(1)A multisection continuum arm based on four pneumatic muscles driven in parallel is self-developed, whose comprehensive kinematic model based on the segmental constant curvature assumption is proposed.(2)A dynamic controller is designed for the condition of unknown parameters of the dynamic model. A strictly double-closed-loop force-position hybrid control strategy is presented for the lumped disturbances of the model, in which adaptive ILC is applied to the inner loop of the controller without a priori knowledge of the dynamic parameters, and the condition of zero initial tracking error is released by employing the trajectory reconstruction method. Moreover, the adaptive approach is implemented to enhance the system’s robustness.(3)The log-type BLF is proposed to always satisfy the time-varying constraint of position-tracking error in the time domain and iterative domain, so as to ensure the safe operation of the pneumatic muscle in the given operating range.(4)The designed system is simulated jointly in an ANSYS/ADAMS/MATLAB environment. Compared with PID control and iterative learning sliding mode control (ILSMC), the effectiveness of the proposed algorithm is demonstrated.

The remainder of this paper is organized as follows: The PMA-based kinematic model of multisection continuum arms is shown in Section 2. The design process of the impedance ILC controller with log-type BLF is given in Section 3. The convergence property of the designed algorithm is demonstrated by means of the composite energy function (CEF) method in Section 4. In Section 5, cosimulation results on ANSYS/ADAMS/MATLAB are depicted to verify the flexible interaction and tracking performance of the proposed algorithm. Finally, conclusions are drawn in Section 6.

## 2. System Modeling and Problem Formulation

A self-developed PMA-based multisection continuum arm is designed as the research object in this paper, whose structure is shown in Figure 1a. The arms are composed of two pneumatic continuum nodes and a rigid connector where each node consists of four pneumatic muscles in parallel (uniformly distributed at 90°), as depicted in Figure 1b. Each pneumatic muscle is connected to the spacer plate by a ball hinge to guarantee that each pneumatic node has four degrees of freedom. The degrees of freedom of a single pneumatic node are calculated as Equation (1). By controlling the magnitude of air pressure in the pneumatic muscles, the individual PMA can be deformed by elongation or bending. What is more, is that the continuum arms are allowed to achieve complicated deformation movements by combining two pneumatic nodes.
(1)$$DOF=6Μ-\sum_{i=1}^{Ν}{Ρ}_{i}$$
where DOF denotes the DOF of a single continuum node, Μ is the number of moving parts of the continuum node, Ν indicates the number of kinematic pairs, ${Ρ}_{i}$ is the number of DOF restricted by the ith kinematic pair. As shown in Figure 1b, Μ=5 and the kinematic pair 1–4 restricts five DOF respectively, and the kinematic pair five restricts sux DOF, so that a single continuum node has four DOF.

### 2.1. Kinematic Modeling Based on Constant Curvature Assumption

For the continuum arms to be precisely controlled, an accurate, stable, and efficient kinematic model is essential. A kinematic model is capable of avoiding nonlinear morphological mapping in joint space. For continuum arms, the relationship between PMA length and robot-end poses is described by the kinematic model. The kinematic modeling process for a multisection continuum arm is presented as follows.
(2)$$X=f_{D-H}\left(f_{b}\left(f_{a}\left(l_{i}\right)\right)\right)$$
where $f_{a}$ denotes the mapping from PMA length to arc parameters; $f_{b}$ is the transformation of arc parameters into the function of discrete joints; $f_{D-H}$ denotes the curve parameter homogeneous transformation matrix (CPHTM) established using the modified D-H method.

#### 2.1.1. Forward Kinematics

The curvature of each section uniform along the centerline of a single pneumatic node requires the following two assumptions: (1) The continuum robot consists of a series of consecutive sections; (2) the potential energy inside each node is uniformly distributed; and the combined force inside each node is uniformly distributed along the centerline of the node. The continuum arms designed in this paper meet the above premises.

The deformation model of a single continuum node is depicted in Figure 2, which can be described by the three arc parameters under the assumption of a constant curvature, bending angle $\theta_{i}$, centerline curvature $k_{i}$ (or radius of curvature $r_{i}$) of the continuum node, and the angle relative to the bending plane concerning the $+X_{i}$ axis $\phi_{i}$. The multisection continuum arms designed in this paper contain two continuum nodes, so i=2. Given the length of the pneumatic muscle $\left(l_{i1},l_{i2},l_{i3},l_{i4}\right)$ in a single continuum node, the distance from the center of the pneumatic muscle to the center of the spacer plate is d. The bottom of the continuum node is denoted as $\left\{O_{i}\right\}$, and the top of the continuum node is denoted as $\left\{O_{i+1}\right\}$. The continuum node can be considered as an infinite number of sheets spliced together, and a single sheet is denoted as $\left\{O_{i}^{\kappa }\right\}$, $\kappa \in \left[0,1\right]$. When $\kappa_{i}=0$, the coordinate $\left(X_{i}^{\kappa },Y_{i}^{\kappa },Z_{i}^{\kappa }\right)$ of sheet $\left\{O_{i}^{\kappa }\right\}$ coincides with the base coordinate $\left(X_{1},Y_{1},Z_{1}\right)$; when $\kappa_{i}=1$, the coordinate $\left(X_{i}^{\kappa },Y_{i}^{\kappa },Z_{i}^{\kappa }\right)$ of sheet $\left\{O_{i}^{\kappa }\right\}$ coincides with the top coordinate $\left(X_{2},Y_{2},Z_{2}\right)$. The arc parameter $\left(\theta_{i},k_{i},\phi_{i}\right)$ of the deformation of the continuum node can be calculated with the geometric method.

A schematic diagram of a single continuum node bending at deflection angle $\phi_{i}$ is provided in Figure 3a, while the top view of the continuum node is shown in Figure 3b. Separate reference coordinate systems are established for the bottom and top. From the geometry of the figure, the radius of curvature measured from the center of the continuum node is related to the radius of curvature of each PMA. In the case of PMA1, the geometric relationship between $r_{i}$ and $r_{1}$ is presented as:(3)$$r_{1}=r_{i}-d\cos\alpha_{1}$$
where $\alpha_{1}$ is the angle between the bending direction of a continuum node and PMA1.

The arc length of the centerline of the continuum node is defined as $l_{i}$. From the arc length formula, $l_{i}=\theta_{i}r_{i}$, $l_{1}=\theta_{i}r_{1}$ is obtained. Then, the arc length formula is substituted into Equation (3) to obtain the relationship between the arc length of the centerline $l_{i}$ and the arc length of the PMA $l_{1}$ (applicable to all PMAs).
(4)$$l_{i}=l_{1}+\theta_{i}d\cos\alpha_{1}$$

In Figure 3b, it is easy to see that the following relationship exists between α and $\phi_{i}$ for each PMA.
(5)$$\left\{\begin{array}{c} \cos\alpha_{1}=\cos(90{}^{\circ}-\phi_{i})=\sin\phi_{i}\\ \cos\alpha_{2}=\cos(180{}^{\circ}-\phi_{i})=-\cos\phi_{i}\\ \cos\alpha_{3}=\cos(270{}^{\circ}-\phi_{i})=-\sin\phi_{i}\\ \cos\alpha_{4}=\cos(360{}^{\circ}-\phi_{i})=\cos\phi_{i} \end{array}\right.$$

Substitute Equation (5) into Equation (4) to get:(6)$$l_{i}(\vec{q})=\frac{l_{1}+l_{2}+l_{3}+l_{4}}{4}$$
where $\vec{q}=\left[l_{1},l_{2},l_{3},l_{4}\right]$,$\sum_{j}^{4}\cos\alpha_{j}=0$.

Generalizing Equation (4) for each PMA and taking the difference of nonadjacent PMAs, Equation (4) can be described as:(7)$$\left\{\begin{array}{c} \theta_{i}d=\frac{l_{3}-l_{1}}{\cos\alpha_{1}-\cos\alpha_{3}}=\frac{l_{3}-l_{1}}{2\sin\phi_{i}}\\ \theta_{i}d=\frac{l_{4}-l_{2}}{\cos\alpha_{2}-\cos\alpha_{4}}=\frac{l_{4}-l_{2}}{-2\cos\phi_{i}} \end{array}\right.$$

Divide the above formula to get the deflection angle $\phi_{i}$.
(8)$$\phi_{i}\left(\vec{q}\right)=arctg(\frac{l_{3}-l_{1}}{l_{2}-l_{4}})$$

Substitute the curvature formula into Equation (3) to obtain centerline curvature $k_{i}$.
(9)$$k_{i}=\frac{l_{i}-l_{ij}}{l_{i}d\cos\alpha_{ij}},j=1,2,3,4$$

Let j=1, substitute Equations (5) and (8) into (9) to get the centerline curvature $k_{i}$ and radius of curvature $r_{i}$.
(10)$$k_{i}(\vec{q})=\frac{\left(-3l_{1}+l_{2}+l_{3}+l_{4}\right)\sqrt{{\left(l_{4}-l_{2}\right)}^{2}+{\left(l_{3}-l_{1}\right)}^{2}}}{d\left(l_{1}+l_{2}+l_{3}+l_{4}\right)\left(l_{3}-l_{1}\right)}$$
(11)$$r_{i}(\vec{q})=\frac{d\left(l_{1}+l_{2}+l_{3}+l_{4}\right)\left(l_{3}-l_{1}\right)}{\left(-3l_{1}+l_{2}+l_{3}+l_{4}\right)\sqrt{{\left(l_{4}-l_{2}\right)}^{2}+{\left(l_{3}-l_{1}\right)}^{2}}}$$

**Remark** **1.**

$\sin(arctg(\frac{l_{3}-l_{1}}{l_{2}-l_{4}}))$

*is involved in the simplification process of the above equation, and the auxiliary triangle can be designed to obtain the following:*

(12)
$$\sin(arctg(\frac{l_{3}-l_{1}}{l_{2}-l_{4}}))=\frac{\sqrt{{\left(l_{4}-l_{2}\right)}^{2}+{\left(l_{3}-l_{1}\right)}^{2}}}{\left(l_{3}-l_{1}\right)}$$



The bending angle $\theta_{i}$ of a continuum node is obtained from $l_{i}=\theta_{i}r_{i}$
(13)$$\theta_{i}(\vec{q})=\frac{\left(-3l_{1}+l_{2}+l_{3}+l_{4}\right)\sqrt{{\left(l_{4}-l_{2}\right)}^{2}+{\left(l_{3}-l_{1}\right)}^{2}}}{4d\left(l_{3}-l_{1}\right)}$$

So far, the functional relationship between all arc parameters of a single continuum node and each PMA has been obtained.

Based on the assumption of equal circular arcs and the structural characteristics of continuum arms, the arc parameters $\left(\theta_{i},k_{i},\phi_{i}\right)$ are utilized to replace the translation parameters D and rotation parameters θ of the traditional robot arm. A single continuum node can be discretized into five joints, as shown in Figure 4. Thus, the arc-parameter homogeneous transition matrix (CPHTM) of the first continuum node is obtained as follows:
(14)$$\begin{array}{l} T_{i}(\vec{q})=T_{rz}\left(\phi_{i}\right)T_{px}\left(r_{i}\right)T_{ry}\left(\theta_{i}\right)T_{px}\left(-r_{i}\right)T_{rz}\left(-\phi_{i}\right)\\ \;=\left[\begin{array}{cc} R_{i}(\vec{q}) & P_{i}(\vec{q})\\ 0_{1\times 3} & 1 \end{array}\right] \end{array}$$
where $T_{rz}\in R^{4\times 4}$ and $T_{ry}\in R^{4\times 4}$ are the rotational homogeneous transformation matrix (RHTM) about the *z*-axis and the *y*-axis, respectively. $T_{px}\in R^{4\times 4}$ is the translation homogeneous transformation matrix (THTM) along the *x*-axis, which is expressed as follows:(15)$$T_{rz}\left(\phi_{i}\right)=\left[\begin{array}{cccc} \cos\phi_{i} & -\sin\phi_{i} & 0 & 0\\ \sin\phi_{i} & \cos\phi_{i} & 0 & 0\\ 0 & 0 & 1 & 0\\ 0 & 0 & 0 & 1 \end{array}\right]$$
(16)$$T_{ry}\left(\theta_{i}\right)=\left[\begin{array}{cccc} \cos\theta_{i} & -\sin\theta_{i} & 0 & 0\\ \sin\theta_{i} & \cos\theta_{i} & 0 & 0\\ 0 & 0 & 1 & 0\\ 0 & 0 & 0 & 1 \end{array}\right]$$
(17)$$T_{px}\left(r_{i}\right)=\left[\begin{array}{cccc} 1 & 0 & 0 & r_{i}\\ 0 & 1 & 0 & 0\\ 0 & 0 & 1 & 0\\ 0 & 0 & 0 & 1 \end{array}\right]$$

$R_{i}(\vec{q})\in SO(3)$ denotes the rotation matrix under the radian parameter, and $P_{i}(\vec{q})\in R^{3\times 1}$ represents the displacement vector under the radian parameter.

**Remark** **2.***Note that the arc parameter expression*$\left(\theta_{i},k_{i},\phi_{i}\right)$*calculated above has singular values, that is, the continuum arms have no numerical solution to Equations (8), (10), and (13) in the vertical state (*$l_{4i}=l_{4i-1}=l_{4i-2}=l_{4i-3}$*). In order to avoid the singular value of the arms in the forward kinematics solution process, the arms are divided into a bending state and a vertical state. When the continuum arms merely stretch vertically in the z-axis,*$R_{i}(\vec{q})$*changes into the unit matrix,*$P_{i}(\vec{q})={\left[0,0,l_{i}\right]}^{T}$.

**Remark** **3.***The multisection continuum arms designed in this paper include a rigid connector where the kinematic model also needs to be considered. As shown in*Figure 5*, the CPHTM of the first continuum node is denoted as*$T\left({\vec{q}}_{1},\kappa_{1}\right)$*, the CPHTM of the second node of the intermediate rigid connector is denoted as*$T_{m}$*, and the CPHTM of the third continuum node is represented as*$T\left({\vec{q}}_{2},\kappa_{2}\right)$*. The intermediate rigid connector is essentially a rigid parallel platform. It is assumed that the rigid connector does not deform during the movement, that is, the RHTM does not change, and only the THTM is calculated*.

The modal homogeneous transformation matrix (MHTM) of the multisection continuum arm is denoted as:(18)T03=Tq→1,κ1Trz0TmTrz0Tq→2,κ2=R¯3×3q→2,κ2P¯3×1q→2,κ201×31

From Equation (18), ${\bar{R}}_{3\times 3}\left({\vec{q}}^{2},\kappa^{2}\right)$ is the modal rotation transformation matrix (MRTM), ${\bar{R}}_{3\times 3}=R_{1}R_{m}R_{2}\in SO(3),R_{m}=1$; ${\bar{P}}_{3\times 1}=P_{1}+P_{m}+R_{1}\left(P_{1}+P_{m}\right)\in R^{3\times 1}$ denotes the modal translation transformation matrix (MTTM); ${\vec{q}}^{2}={\left[{\vec{q}}_{1},{\vec{q}}_{2}\right]}^{T}\in R^{2\times 4}$ represents the vector set of joint space, ${\vec{q}}^{i}={\left[q_{4i-3},q_{4i-2},q_{4i-1},q_{4i}\right]}^{T}$; $\kappa^{2}$ indicates the vector set of slice scalar coefficients, $\kappa^{2}={\left[\kappa_{1},\kappa_{2}\right]}^{T}\in R^{2}$; The installation position has 0° deviation around the z-axis. Please refer to Appendix A for specific values.

#### 2.1.2. Inverse Kinematics

In the previous subsection, the MHTM of a multisection continuum arm is calculated given the known length of the PMA. The lengths of eight PMAs are calculated in this section given the end poses of the arms. In order to facilitate the calculation and implementation, the inverse kinematics model of continuum arms is studied with the method of geometry.
(19)$$l_{i}=f_{a}^{*}\left(f_{b}^{*}\left(X\right)\right)$$

In the above formula, $f_{b}^{*}$ represents the functional relationship of transforming the arms end pose into arc parameters; $f_{a}^{*}$ denotes the functional relationship from arc parameters to PMA length.

Taking a single continuum node as an example for analysis, the pose matrix of the top coordinate system $\left(X_{2},Y_{2},Z_{2}\right)$ relative to the base coordinate system $\left(X_{1},Y_{1},Z_{1}\right)$ is:(20)T01=rxxrxyrxzpxryxryyryzpyrzxrzyrzzpz0001

The value of $p_{x}$,$p_{y}$,$p_{z}$ in Equation (20) needs to be given in advance. In the first step, the functional relationship between the pose matrix and the arc parameters is to be established, as depicted in Figure 6. Drawing upon the knowledge of geometry, the functional relationship can be expressed by the following equation:(21)$$\phi \left(\vec{q}\right)=arctg\frac{y_{2}}{x_{2}}$$
(22)$$\theta \left(\vec{q}\right)=2*arctg\left(\frac{x_{2}}{z_{2}*\cos\phi }\right)$$
(23)$$r\left(\vec{q}\right)=\frac{z_{2}}{\sin\theta }$$

In the second step, the functional relationship between the arc parameters and the length of each PMA is to be determined. In combination with Figure 6, an auxiliary plane N is designed, which passes through the lowest point M at the top of the node $\left\{O_{i+1}\right\}$ and is perpendicular to each PMA, see Figure 7a. A straight line is drawn through the point $A_{1}$,$A_{2}$,$A_{3}$,$A_{4}$,O, which is perpendicular to the plane N, and the height of $A_{1}$,$A_{2}$,$A_{3}$,$A_{4}$,O from the plane N is denoted as $h_{1\phi }$,$h_{2\phi }$,$h_{3\phi }$,$h_{4\phi }$,$h_{O\phi }$, as shown in Figure 7b.

Combined with Figure 6, the height from the center of the circle to the plane N is obtained as follows:(24)$$h_{O\phi }=d\sin\left(\frac{\theta }{2}\right)$$

It can be seen from Figure 7b that $h_{1\phi }$,$h_{2\phi }$,$h_{3\phi }$,$h_{4\phi }$,$h_{O\phi }$ and $A_{1}$,$A_{2}$,$A_{3}$,$A_{4}$,O are proportional to the distance $d_{x}$ from point $M_{b}$ in the projected view.
(25)$$\frac{h_{x\phi }}{h_{O\phi }}=\frac{d_{x}}{d}$$

From Figure 7c, the distances from point $A_{1}$,$A_{2}$,$A_{3}$,$A_{4}$ to the straight line $M_{t}M_{b}$ in the projected view can be expressed as the following equation:(26)$$\left\{\begin{array}{c} d_{1}=d\left(1-\cos\phi \right)\\ d_{2}=d\left(1+\sin\phi \right)\\ d_{3}=d\left(1+\cos\phi \right)\\ d_{4}=d\left(1-\sin\phi \right) \end{array}\right.$$

Hence,
(27)$$\left\{\begin{array}{c} h_{1\phi }=d\left(1-\cos\phi \right)\sin\left(\frac{\theta }{2}\right)\\ h_{2\phi }=d\left(1+\sin\phi \right)\sin\left(\frac{\theta }{2}\right)\\ h_{3\phi }=d\left(1+\cos\phi \right)\sin\left(\frac{\theta }{2}\right)\\ h_{4\phi }=d\left(1-\sin\phi \right)\sin\left(\frac{\theta }{2}\right) \end{array}\right.$$

The chord length from the base $\left\{O_{1}\right\}$ to the plane N in the bending plane is defined as $\chi_{\phi }$, $\chi_{\phi }$ and can be denoted as:(28)$$\chi_{\phi }=2\sin\left(\frac{\theta }{2}\right)\left(r-d\right)$$

From the definition of plane N, the relationship between the chord lengths $h_{1\phi }$,$h_{2\phi }$,$h_{3\phi }$,$h_{4\phi }$, and $\ell_{1}$,$\ell_{2}$,$\ell_{3}$,$\ell_{4}$ of PMA can be obtained as:(29)$$\left\{\begin{array}{c} \ell_{1}=\chi_{\phi }+2h_{1\phi }=2\sin\left(\frac{\theta }{2}\right)\left(r-d\cos\phi \right)\\ \ell_{2}=\chi_{\phi }+2h_{2\phi }=2\sin\left(\frac{\theta }{2}\right)\left(r-d\sin\phi \right)\\ \ell_{3}=\chi_{\phi }+2h_{3\phi }=2\sin\left(\frac{\theta }{2}\right)\left(r+d\cos\phi \right)\\ \ell_{4}=\chi_{\phi }+2h_{4\phi }=2\sin\left(\frac{\theta }{2}\right)\left(r+d\sin\phi \right) \end{array}\right.$$

The relationship between the chord length of PMA and the length of each PMA is:(30)$$l_{j}=\frac{\ell_{j}\theta }{2\sin\frac{\theta }{2}}$$

Eventually, the length of each PMA is obtained as:(31)$$\left\{\begin{array}{c} l_{1}=\theta r-\theta d\cos\phi \\ l_{2}=\theta r-\theta d\sin\phi \\ l_{3}=\theta r+\theta d\cos\phi \\ l_{4}=\theta r+\theta d\sin\phi \end{array}\right.$$

The inverse solution process of the multisection continuum arms is basically the same as the single continuum node, and will not be repeated in this paper.

**Remark** **4.**
*Note that Equation (31) is consistent with Equations (4) and (5). Nonetheless, two geometric methods to model the forward and inverse kinematics of the multisection continuum arms are proposed in this paper.*


The kinematic modeling errors are presented in Table 1. A total of 50 groups of experiments were carried out in this paper, and some of the results were selected for display. Compared with the ideal length of PMA, the actual length of PMA is obtained through the inverse kinematics solution. It can be intuitively concluded that the kinematic modeling accuracy described in this paper can reach 99.987% where the validity of the kinematic model is verified by experiments.

### 2.2. Dynamic Model with Unknown Parameters

It is indispensable for the continuum arms to interact with the external environment in practical applications. In more detail, this interaction is not only reflected at the cognitive level (information transfer), but also at the physical level (interaction torque). The arms’ movement and control stability will be affected by the interaction torque generated by contact. Therefore, the main disturbance types are analyzed in the whole continuum arms, and on this basis, the dynamic model of multisection continuum arms with unknown model parameters is established.

The uncertainties in the continuum arms mainly include external disturbances, interaction torques, model parameter errors and dynamic changes. $d_{ex}(t)$ is defined as the external disturbance and model error of the system. $d_{em}(t)$ is represented as the interaction torque generated in the robot space which can be expressed as:(32)$$d_{em}(t)=J^{T}F_{\mathrm{int}}(t)$$
where $J^{T}$ is the Jacobian matrix, and $F_{\mathrm{int}}(t)$ is the force generated by the external environment on the end of the robot, that is, the interaction force.

**Remark** **5.***Considering the periodic motion of the ends of the continuum arms, the model parameter error and the interaction torque also exhibit periodic characteristics, which can be regarded as a part of the ideal model parameters obtained by iterative learning. Therefore, the specific value of the Jacobian matrix is not solved in this paper. According to the above analysis, the lumped disturbance is represented by*$d_{lump}(t)$ *of the continuum arms. Ultimately, the dynamic equation of the multisection continuum arms under the condition of unknown model parameters can be expressed as:*(33)$$M_{x}(q)\ddot{q}(t)+C_{x}(q,\dot{q})\dot{q}(t)+G_{x}(q)+d_{lump}(t)=u(t)$$
where q(t),$\dot{q}(t)$,and $\ddot{q}(t)$ denote the state vectors of position, velocity, and acceleration of each PMA. $u(t)\in R^{8\times 1}$ indicates the output torque of each PMA. $M_{x}(q)\in R^{8\times 8}$ denotes the inertial matrix. $C_{x}(q,\dot{q})\in R^{8\times 8}$ represents the Coriolis/centrifugal matrix. $G_{x}(q)\in R^{8\times 1}$ indicates the gravity vector.

**Remark** **6.**
*The position vector set of the PMA in the previous section is represented by*

${\vec{q}}_{i}(t)={\left[l_{1}\left(t\right),l_{2}\left(t\right),l_{3}\left(t\right),l_{4}\left(t\right)\right]}^{T},i=1,2$

*. For the convenience of expression in the following text,*

$q(t)\in R^{8\times 1}$

*is uniformly applied to represent the position matrix of the PMA.*


The proposed dynamic equation has the following properties:

**Property** **1.***The inertia matrix*$M_{x}(q)\in R^{8\times 8}$*is a symmetric positive definite matrix which exists in the upper and lower bounds*.
(34)$$0<\delta_{\min}\left(M_{x}\right)\leq \left\|M_{x}(q)\right\|\leq \delta_{\max}\left(M_{x}\right)$$
where $\delta_{\min}\left(M_{x}\right)$ is the smallest eigenvalue of the positive definite matrix $M_{x}$; $\delta_{\max}\left(M_{x}\right)$ denotes the largest eigenvalue of the positive definite matrix $M_{x}$. $\left\|\cdot \right\|$ stands for the Euclidean norm operation.

**Property** **2.**${\dot{M}}_{x}(q)-2C_{x}(q,\dot{q})$*is an obliquely symmetric matrix*.
(35)$$\sigma^{T}\left[{\dot{M}}_{x}(q)-2C_{x}(q,\dot{q})\right]\sigma =0$$

**Property** **3.***The gravity matrix has an upper bound*.
(36)$$G_{x}(q)\leq {ο}_{\max}$$

For the convenience of subsequent expressions, Formula (33) is rewritten as:(37)$$\left\{\begin{array}{l} {\dot{x}}_{1}(t)=x_{2}(t)\\ {\dot{x}}_{2}(t)=-M_{x}^{-1}\left(x_{1}\right)\left(C_{x}(x)x_{2}(t)+G_{x}\left(x_{1}\right)+d_{lump}(t)\right)+M_{x}^{-1}\left(x_{1}\right)u(t) \end{array}\right.$$
where the state variable is $x_{1}(t)=q(t),x_{2}(t)=\dot{q}(t)$.

The expected position trajectories of the eight PMAs at the kth iteration are denoted as $x_{ck}(t)\in R^{8\times 1}$. It should be noted that, to obtain the desired motion trajectory $x_{ck}(t)$ in the joint space, the position-based impedance control is implemented to correct the trajectory of the robot end. Then, the inverse kinematics solution is utilized to obtain $x_{ck}(t)$. Then, the position and velocity tracking errors of each iteration can be expressed as:(38)$$s_{1k}(t)=x_{ck}(t)-x_{1k}(t),{\dot{s}}_{1k}(t)={\dot{x}}_{ck}(t)-x_{2k}(t)$$

**Remark** **7.**
*For the convenience of the following description,*

$M_{x}(q)$

*,*

$C_{x}(q,\dot{q})$

*,*

$G_{x}(q)$

*,*

$d_{lump}(t)$

*is replaced by*

$M_{xk}$

*,*

$C_{xk}$

*,*

$G_{xk}$

*,*

$d_{lump,k}(t)$

*, respectively.*


### 2.3. Problem Formulation

The position-tracking control problem studied can be expressed as follows: The control law u(t) is designed to guarantee that the position-tracking error of each PMA can stably converge to 0 and not exceed the given time-varying constraint function ψ(t) under the influence of unknown model parameters and external lumped disturbances. Hence, the control requirement can be described as:(39)$$\underset{t\to \infty }{\lim}\left\|s_{1k}(t)\right\|=0,\forall t\in \left[0,+\infty \right)\;\&\;\left\|s_{1k}(t)\right\|\leq \psi (t),\left\|s_{1k}(t)\right\|\in R^{8\times 1}$$

Moreover, as for ensuring that the ideal static balance force (ISBF) $F_{id}$ remains constant when the arms interact with the external environment, a position-based impedance control strategy is investigated, and the force control requirements are achieved by correcting the expected trajectory in the Cartesian space.
(40)$$F_{id}=f_{con}$$
where $f_{con}$ is a constant.

## 3. Controller Design

The dynamic equation of the multisection continuum arms has been presented in the previous section. The proposed arms are characterized by high nonlinearity, strong time-varying coupling, creep, and uncertainty, which cause the controller design to be extremely difficult under the existing conditions. In this paper, a double closed-loop control scheme is proposed for a multisection continuum arm driven by eight PMAs. For one thing, the impedance control acts as the outer loop of the controller to ensure that the ISBF stays constant. For another, adaptive ILC is combined with backstepping control as the inner loop of the proposed controller. Specifically, the ILC is employed to estimate the unknown time-varying parameters of the dynamic equation, and the adaptive algorithm is implemented to suppress the upper bound of the lumped disturbance and feed it back to the controller for real-time compensation to enhance the robustness of the continuum arms. Meanwhile, as to guarantee that the position-tracking errors of the arms converge within the given time-varying constraint function, a backstepping control strategy based on the log-type BLF is designed. The control flow chart is depicted in Figure 8.

In order to fulfill the desired control requirement, the following assumptions and lemmas are given first, which facility the controller design in subsequent discussion.

**Assumption** **1.***The desired trajectory*$x_{ck}(t)$*of the joint space is continuously bounded and second-order differentiable, i.e.,*${\ddot{x}}_{ck}(t)$*exists.*$\zeta (t)={\left[M_{xk},C_{xk},G_{xk}\right]}^{T}$*is defined as the unknown parameter matrix. For a given desired trajectory*$x_{ck}(t)$*, there exists an ideal parameter*$\zeta_{d}={\left[M_{xd},C_{xd},G_{xd}\right]}^{T}$*, such that the actual trajectory of each PMA can track the desired trajectory*.

**Assumption** **2.***The lumped disturbance*$d_{lump,k}(t)$*is a slow time-varying disturbance and is uniformly bounded, i.e.,*${\dot{d}}_{lump,k}(t)\approx 0$*and*$\left\|d_{lump,k}(t)\right\|\leq D_{\max}$.

**Lemma** **1.***The following equations are valid for arbitrary matrices*$U,V,W\in R^{m\times n}$.
(41)$$tr\left[{\left(U-V\right)}^{T}\left(U-V\right)-{\left(U-W\right)}^{T}\left(U-W\right)\right]=tr\left[{\left(W-V\right)}^{T}\left(2\left(U-V\right)+\left(V-W\right)\right)\right]$$
where the trace of the matrix is represented by $tr\left(\cdot \right)$.

**Lemma** **2.***For arbitrary functions* $\psi_{1}(t)$ *and*$\psi_{2}(t)$*, let* $N:=R^{l}\times Z_{1}\subset R^{l+1}$ *and* $Z_{1}:=\left\{z_{1}\in R:-\psi_{1}(t)<z_{1}<\psi_{2}(t)\right\}\subset R$ *be open intervals for the system.*(42)$$\dot{\eta }=h\left(t,\eta \right)$$
where $\eta ={\left[w,z_{1}\right]}^{T}\in N$, and $C:R^{+}\times N\to R^{l+1}$ are piecewise continuous functions and satisfy the global Lipschitz condition with respect to $z_{1}$ in the $R^{+}\times N$ domain. If there exist continuously differentiable positive definite functions $Q:R^{l}\to R^{+}$ and $V_{1}:Z_{1}\to R^{+}$ satisfy the following:(43)$$z_{1}\to -\psi_{1}(t)\cup z_{1}\to \psi_{2}(t),V_{1}\left(z_{1}\right)\to \infty$$
(44)$$\underline{\alpha }\left(\left\|w\right\|\right)\leq Q\left(w\right)\leq \bar{\alpha }\left(\left\|w\right\|\right)$$
where $\underline{\alpha }$ and $\bar{\alpha }$ are $k_{\infty }$ functions. Define $V\left(\eta \right)=V_{1}\left(z_{1}\right)+Q\left(w\right)$, $z_{1}\left(0\right)$ is in the open set $\left(-\psi_{1}(t),\psi_{2}(t)\right)$, if
(45)$$\dot{V}=\frac{\partial V}{\partial \eta }h\leq 0$$

Then, $z_{1}$ will always remain in the open set $z_{1}\in \left(-\psi_{1}(t),\psi_{2}(t)\right)$.

**Lemma** **3.***For any* $\left\|s_{1k}(t)\right\|\leq \psi (t)$*, the following inequality holds:*(46)$$\log\frac{\psi^{2}(t)}{\psi^{2}(t)-s_{1k}^{T}(t)s_{1k}(t)}\leq \frac{s_{1k}^{T}(t)s_{1k}(t)}{\psi^{2}(t)-s_{1k}^{T}(t)s_{1k}(t)}$$

### 3.1. Impedance Control Strategy

In this paper, the position-based impedance control strategy is applied to obtain the position deviation under the change of the interaction force so that the desired trajectory is corrected. The equation expression is shown as follows:(47)$$F_{id}-F_{\mathrm{int}}(t)=H\left({\ddot{x}}_{d}(t)-{\ddot{x}}_{r}(t)\right)+B\left({\dot{x}}_{d}(t)-{\dot{x}}_{r}(t)\right)+K\left(x_{d}(t)-x_{r}(t)\right)$$
where $F_{id}$ denotes the ISBF, H,B,K, respectively indicates the inertia coefficient, damping coefficient and stiffness coefficient. $x_{d}(t),x_{r}(t)$ represents the command trajectory and desired trajectory in the Cartesian space.

Equation (47) is converted to the transfer function form.
(48)$$x_{e}(t)=\frac{F_{id}-F_{\mathrm{int}\;}(t)}{Hs^{2}+Bs+K}$$
where $x_{e}(t)=x_{d}(t)-x_{r}(t)$ is the adjustment amount of the command trajectory.

According to the adjustment amount of the command trajectory; the desired trajectory of the continuum arms is obtained as:(49)$$x_{r}(t)=x_{d}(t)-x_{e}(t)=x_{d}(t)-\frac{F_{id}-F_{\mathrm{int}}(t)}{Hs^{2}+Bs+K}$$

**Remark** **8.**
*It can be concluded from Equation (49) that when the interaction force is greater than the ISBF, i.e.,*

$F_{\mathrm{int}}(t)>F_{id}$

*, on account of*

H,B,K>0

*, the adjustment amount of the command trajectory is obtained as*

$x_{e}(t)<0$

*, i.e.,*

$x_{d}(t)<x_{r}(t)$

*. For instance, the end position of the desired trajectory becomes larger than the end position of the command trajectory while the multisection continuum arms accelerate to stretch or decelerate to contract, which causes the interaction force to decrease, and vice versa.*


Eventually, the inverse kinematics solution (2.1.2) is implemented to obtain the desired motion trajectory $x_{ck}(t)$ of each PMA in the joint space, which is appointed to the tracking target of the position controller.

### 3.2. Adaptive Iterative Learning Backstepping Control Strategy with Initial Error

#### 3.2.1. Reconstruction of the Desired Trajectory

The initial value of the system state is strictly required by traditional ILC, which ensures that the initial state of the system at each iteration is strictly consistent with the initial state of the desired trajectory, i.e., $x_{ck}(0)=x_{1k}(0)=x_{10}(0)$. Nevertheless, in practical applications, the initial state of the continuum arms and the initial state of the desired trajectory are hardly guaranteed to be equal in the iterative domain. Hence, as for the aforementioned phenomenon, the initial condition of ILC is released by the trajectory reconstruction approach when a fixed deviation permanently exists between the initial state of the system and the initial state of the desired trajectory.

The main idea of the trajectory reconstruction approach is that the initial value of the reconstructed trajectory is equal to the initial value of the state in the trajectory operating interval $\left[0,T\right]$. Meanwhile, the reconstructed trajectory at time $\left[\rho ,T\right]$ is the same as the original desired trajectory. The reconstructed trajectory form is shown as follows:(50)$$x_{gk}(t)=x_{ck}(t)-\Gamma_{1}(t)\left(x_{ck}(0)-x_{1k}(0)\right)-\Gamma_{2}(t)\left({\dot{x}}_{ck}(0)-x_{2k}(0)\right)$$
where $\Gamma_{1}(t),\Gamma_{2}(t)$ is the time-varying function which is not unique. However, the following conditions are required to be satisfied when the time-varying function is designed.
(51)$$\left\{\begin{array}{l} x_{gk}(0)=x_{1k}(0)\\ x_{gk}\left(\rho \right)=x_{ck}\left(\rho \right)\\ x_{gk}^{\left(m\right)}\left(\rho \right)=x_{ck}^{\left(m\right)}\left(\rho \right) \end{array}\right.$$
${\left(\cdot \right)}^{\left(m\right)}$ represents the m-order derivative of the function, and m≥2 in this paper.

The time-varying function Γ(t) can be designed as:(52)$$\Gamma_{1}(t)=\left\{\begin{array}{l} {\left(1-\frac{t}{\rho }\right)}^{3},\;t\in \left[0,\rho \right)\\ 0,\;t\in \left[\rho ,T\right] \end{array}\right.$$
(53)$$\Gamma_{2}(t)=\left\{\begin{array}{l} t*{\left(1-\frac{t}{\rho }\right)}^{3},\;t\in \left[0,\rho \right)\\ 0,\;t\in \left[\rho ,T\right] \end{array}\right.$$

#### 3.2.2. Adaptive Iterative Learning Backstepping Controller

**Step 1:** To implement $\left\|s_{1k}(t)\right\|\leq \psi (t),\forall t>0$, the following log-type BLF is defined as:(54)$$V_{1k}(t)=\frac{1}{2}\log\frac{\psi^{2}(t)}{\psi^{2}(t)-s_{1k}^{T}(t)s_{1k}(t)}$$

Then, its derivative with respect to time can be expressed as:(55)$${\dot{V}}_{1k}(t)\leq \frac{s_{1k}^{T}(t){\dot{s}}_{1k}(t)-Q_{1}s_{1k}^{T}(t)s_{1k}(t)}{\psi^{2}(t)-s_{1k}^{T}(t)s_{1k}(t)}$$
where $Q_{1}=\sup_{t}\sqrt{\left(\frac{\dot{\psi }(t)}{\psi (t)}\right)+\varepsilon_{k}},\varepsilon_{k}>0$ is the auxiliary variable.

Then, introduce the stable function $z_{1k}(t)$ to be designed, while the position-tracking error and the tracking error of the stable function is defined as:(56)$$s_{1k}(t)=x_{gk}(t)-x_{1k}(t),s_{2k}(t)=z_{1k}(t)-x_{2k}(t)$$

Equation (55) can be expressed as:(57)$${\dot{V}}_{1k}(t)\leq \frac{s_{1k}^{T}(t)\left({\dot{x}}_{gk}(t)+s_{2k}(t)-z_{1k}(t)\right)-Q_{1}s_{1k}^{T}(t)s_{1k}(t)}{\psi^{2}(t)-s_{1k}^{T}(t)s_{1k}(t)}$$

As to ensure the stability of the system, the stability function $z_{1k}(t)$ is designed as:(58)$$z_{1k}(t)={\dot{x}}_{gk}(t)+k_{1}s_{1k}(t)\left(\psi^{2}(t)-s_{1k}^{T}(t)s_{1k}(t)\right)-Q_{1}s_{1k}(t),k_{1}>0$$

Substitute the above formula into ${\dot{V}}_{1k}(t)$
(59)$${\dot{V}}_{1k}(t)\leq -k_{1}s_{1k}^{T}(t)s_{1k}(t)+\frac{s_{1k}^{T}(t)s_{2k}(t)}{\psi^{2}(t)-s_{1k}^{T}(t)s_{1k}(t)}$$

If $s_{2k}(t)=0$, ${\dot{V}}_{1k}(t)\leq -k_{1}s_{1k}^{2}(t)$. Hence, the controller needs to be designed in step 2.

**Step 2:** Since the state variables in Equation (37) do not need to be constrained, the following Lyapunov function is defined as:(60)$$V_{2k}(t)=V_{1k}(t)+\frac{1}{2}s_{2k}^{T}(t)M_{xk}s_{2k}(t)+\frac{1}{2\gamma }{\tilde{d}}_{lump,k}^{T}(t){\tilde{d}}_{lump,k}(t)$$
where γ>0 is the adaptive step size.

The adaptive estimation error of the lumped disturbance is:(61)$${\tilde{d}}_{lump,k}(t)=d_{lump,k}(t)-{\hat{d}}_{lump,k}(t)$$

${\hat{d}}_{lump,k}(t)$ denotes the adaptive estimation value of the lumped disturbance. From Assumption 2, it can be known that the lumped disturbance is a slow time-varying disturbance, hence, ${\dot{\tilde{d}}}_{lump,k}(t)=-{\dot{\hat{d}}}_{lump,k}(t)$. Then, derive Equation (60) with respect to time t.
(62)$$\begin{array}{c} {\dot{V}}_{2k}(t)\leq -k_{1}s_{1k}^{T}(t)s_{1k}(t)+\frac{s_{1k}^{T}(t)s_{2k}(t)}{\psi^{2}(t)-s_{1k}^{T}(t)s_{1k}(t)}+s_{2k}^{T}(t)M_{xk}{\dot{s}}_{2k}(t)\\ +\frac{1}{2}s_{2k}^{T}(t){\dot{M}}_{xk}s_{2k}(t)+\frac{1}{\gamma }{\tilde{d}}_{lump,k}^{T}(t){\dot{\tilde{d}}}_{lump,k}(t)\\ \leq -k_{1}s_{1k}^{T}(t)s_{1k}(t)+s_{2k}^{T}(t)\left[M_{xk}{\dot{z}}_{1k}(t)+C_{xk}x_{2k}(t)+G_{xk}+d_{lump,k}(t)-u(t)\right]\\ \;+\frac{s_{1k}^{T}(t)s_{2k}(t)}{\psi^{2}(t)-s_{1k}^{T}(t)s_{1k}(t)}+\frac{1}{2}s_{2k}^{T}(t){\dot{M}}_{xk}s_{2k}(t)-\frac{1}{\gamma }{\tilde{d}}_{lump,k}^{T}(t){\dot{\hat{d}}}_{lump,k}(t) \end{array}$$


Combined with Property 2 of dynamic Equation (33), it can be concluded that $s_{2k}^{T}(t)\left[{\dot{M}}_{xk}-2C_{xk}\right]s_{2k}(t)=0$. Therefore, the above formula can be simplified to:(63)$$\begin{array}{c} {\dot{V}}_{2k}(t)\leq -k_{1}s_{1k}^{T}(t)s_{1k}(t)+\frac{s_{1k}^{T}(t)s_{2k}(t)}{\psi^{2}(t)-s_{1k}^{T}(t)s_{1k}(t)}-\frac{1}{\gamma }{\tilde{d}}_{lump,k}^{T}(t){\dot{\hat{d}}}_{lump,k}(t)\\ +s_{2k}^{T}(t)\left[\zeta_{d}^{T}(t)\Phi_{k}(t)+d_{lump,k}(t)-u(t)\right] \end{array}$$
where $\zeta_{d}(t)={\left[M_{xk},C_{xk},G_{xk}\right]}^{T}$ represents the ideal unknown parameter matrix. $\Phi_{k}(t)={\left[{\dot{z}}_{1k}^{T}(t),z_{1k}^{T}(t),1\right]}^{T}$ denotes the state matrix of the kth iteration.

Ultimately, the control law is designed as:(64)$$u(t)={\hat{\zeta }}_{k}^{T}(t)\Phi_{k}(t)+{\hat{d}}_{lump,k}(t)+\frac{s_{1k}(t)}{\psi^{2}(t)-s_{1k}^{T}(t)s_{1k}(t)}+k_{2}s_{2k}(t)+k_{3}\mathrm{sgn}\left(s_{2k}(t)\right)$$
where $\mathrm{sgn}\left(s_{2k}(t)\right)$ indicates a sign function. The self-adaptive estimated value of the ideal unknown parameter matrix is defined as ${\hat{\zeta }}_{k}(t)$. With the increase in the number of iterations, the ideal unknown parameter matrix $\zeta_{d}(t)$ is gradually approximated to ${\hat{\zeta }}_{k}(t)$, which has the recursive characteristic. ${\hat{\zeta }}_{k}(t)$ can be expressed by the iterative learning law.
(65)$${\hat{\zeta }}_{k}(t)={\hat{\zeta }}_{k-1}(t)+\varsigma \Phi_{k}(t)s_{2k}^{T}(t)$$
where ς denotes the iterative learning step size.

In Equation (64), ${\hat{d}}_{lump,k}(t)$ represents the self-adaptive estimated value of the lumped disturbance $d_{lump,k}(t)$. The self-adaptive law is designed as:(66)$${\dot{\hat{d}}}_{lump,k}(t)=\gamma s_{2k}(t)$$
where γ denotes the adaptive step size.

Substitute Equation (64) into (62).
(67)$${\dot{V}}_{2k}(t)\leq -k_{1}{\left\|s_{1k}(t)\right\|}^{2}-k_{2}{\left\|s_{2k}(t)\right\|}^{2}-k_{3}\mathrm{sgn}\left(s_{2k}(t)\right)+s_{2k}^{T}(t){\tilde{\zeta }}_{k}^{T}(t)\Phi_{k}(t)$$

The adaptive estimation error of the ideal unknown parameter matrix is proposed as:(68)$${\tilde{\zeta }}_{k}(t)=\zeta_{d}(t)-{\hat{\zeta }}_{k}(t)$$

If the parameter estimation matrix ${\hat{\zeta }}_{k}(t)$ is capable of stably converging to $\zeta_{d}(t)$ in the iterative domain, it can be summarized in Equation (67) that the continuum arms can accurately track the reconstructed desired trajectory $x_{gk}(t)$.

## 4. Convergence Analysis

**Theorem** **1.***In this paper, the control law (64), iterative learning law (65), and adaptive law (66) are designed to solve the trajectory tracking control problem of the multisection continuum arms (33) under the condition of unknown model parameters. As the number of iterations*k*increases, the ideal parameter matrix*$\zeta_{d}(t)$*is approximated to the parameter estimation matrix*${\hat{\zeta }}_{k}(t)$*with arbitrary precision. Moreover, the tracking errors*$s_{1k}(t)$*and*$s_{2k}(t)$*can stably converge to 0 in the time period* $\left[\rho ,T\right]$*, and the tracking error is limited by the time-varying constraint function*ψ(t)*within* ∀t≥0.

**Proof.** To evaluate the convergence of the proposed control scheme in the iterative and time domains, comprehensive energy function (CEF) is constructed as:(69)$$\Omega_{k}(t)=\sum_{i=1}^{4}V_{k}^{i}(k),i=1,2,3,4$$
where
(70)$$V_{k}^{1}(t)=\frac{1}{2}\log\frac{\psi^{2}(t)}{\psi^{2}(t)-s_{1k}^{T}(t)s_{1k}(t)}$$
(71)$$V_{k}^{2}(t)=\frac{1}{2}s_{2k}^{T}(t)M_{xk}s_{2k}(t)$$
(72)$$V_{k}^{3}(t)=\frac{1}{2\gamma }{\tilde{d}}_{lump,k}^{T}(t){\tilde{d}}_{lump,k}(t)$$
(73)$$V_{k}^{4}(t)=\frac{1}{2\varsigma }\int_{0}^{t}tr\left({\tilde{\zeta }}_{k}^{T}(t){\tilde{\zeta }}_{k}(t)\right)□$$

**Remark** **9.***In practical application, the preconditions of Equations (59) and (63) need to be satisfied when the CEF is designed. Moreover, a new type of Lyapunov convergence proof is implemented drawing upon Lemma 1 without a unique approach*.


**Step 1:** Prove that the CEF is bounded.

The difference between the CEF of the kth and the $\left(k-1\right)th$ iteration is obtained as:(74)$$\Delta \Omega_{k}(t)=\Delta V_{k}^{1}(t)+\Delta V_{k}^{2}(t)+\Delta V_{k}^{3}(t)+\Delta V_{k}^{4}(t)$$

The BLF $\Delta V_{k}^{1}(t)$ of the above formula can be expressed as:(75)$$\begin{array}{l} \Delta V_{k}^{1}(t)=\frac{1}{2}\log\frac{\psi^{2}(t)}{\psi^{2}(t)-s_{1k}^{T}(t)s_{1k}(t)}-\frac{1}{2}\log\frac{\psi^{2}(t)}{\psi^{2}(t)-s_{1k-1}^{T}(t)s_{1k-1}(t)}\\ \;\leq \frac{1}{2}\log\frac{\psi^{2}(0)}{\psi^{2}(0)-s_{1k}^{T}(0)s_{1k}(0)}-\frac{1}{2}\log\frac{\psi^{2}(t)}{\psi^{2}(t)-s_{1k-1}^{T}(t)s_{1k-1}(t)}\\ \;+\int_{0}^{t}\left[-k_{1}s_{1k}^{T}(t)s_{1k}(t)+\frac{s_{1k}^{T}(t)s_{2k}(t)}{\psi^{2}(t)-s_{1k}^{T}(t)s_{1k}(t)}\right]d\tau \end{array}$$

$s_{1k}(0)=x_{gk}(0)-x_{1k}(0)=0$ is known from Equation (50). Hence, the above formula can be simplified as follows:(76)$$\Delta V_{k}^{1}(t)\leq -\frac{1}{2}\log\frac{\psi^{2}(t)}{\psi^{2}(t)-s_{1k-1}^{T}(t)s_{1k-1}(t)}+\int_{0}^{t}\left[-k_{1}s_{1k}^{T}(t)s_{1k}(t)+\frac{s_{1k}^{T}(t)s_{2k}(t)}{\psi^{2}(t)-s_{1k}^{T}(t)s_{1k}(t)}\right]d\tau$$

The second term $\Delta V_{k}^{2}(t)$ in equation (74) can be written as:(77)$$\begin{array}{l} \Delta V_{k}^{2}(t)=\frac{1}{2}s_{2k}^{T}(t)M_{xk}s_{2k}(t)-\frac{1}{2}s_{2k-1}^{T}(t)M_{xk-1}s_{2k-1}(t)\\ \;=\frac{1}{2}s_{2k}^{T}(0)M_{xk}s_{2k}(0)-\frac{1}{2}s_{2k-1}^{T}(t)M_{xk-1}s_{2k-1}(t)\\ \;+\int_{0}^{t}s_{2k}^{T}(t)\left[\zeta_{d}^{T}(t)\Phi_{k}(t)+d_{lump,k}(t)-u(t)\right]d\tau \end{array}$$

From formula (58), we have:(78)$$s_{2k}(0)=z_{1k}(0)-x_{2k}(0)={\dot{x}}_{gk}(0)-x_{2k}(0)=0$$

Substituting Equation (78) and the control law into Equation (77), we get:(79)$$\begin{array}{l} \Delta V_{k}^{2}(t)=\int_{0}^{t}s_{2k}^{T}(t){\tilde{\zeta }}_{k}^{T}(t)\Phi_{k}(t)d\tau +\int_{0}^{t}s_{2k}^{T}(t){\tilde{d}}_{lump,k}(t)d\tau -\int_{0}^{t}\frac{s_{2k}^{T}(t)s_{1k}(t)}{\psi^{2}(t)-s_{1k}^{T}(t)s_{1k}(t)}d\tau \\ \;-\frac{1}{2}s_{2k-1}^{T}(t)M_{xk-1}s_{2k-1}(t)-\int_{0}^{t}k_{2}{\left\|s_{2k}(t)\right\|}^{2}d\tau -\int_{0}^{t}k_{3}\left\|s_{2k}(t)\right\|d\tau \end{array}$$

For the third term $\Delta V_{k}^{3}(t)$ in Equation (74), we have:(80)$$\begin{array}{l} \Delta V_{k}^{3}(t)=\frac{1}{2\gamma }{\tilde{d}}_{lump,k}^{T}(t){\tilde{d}}_{lump,k}(t)-\frac{1}{2\gamma }{\tilde{d}}_{lump,k-1}^{T}(t){\tilde{d}}_{lump,k-1}(t)\\ \;=-\frac{1}{\gamma }\int_{0}^{t}{\tilde{d}}_{lump,k}^{T}(t){\dot{\hat{d}}}_{lump,k}(t)d\tau -\frac{1}{2\gamma }{\tilde{d}}_{lump,k-1}^{T}(t){\tilde{d}}_{lump,k-1}(t) \end{array}$$

Equation (80) is substituted into the adaptive law (66), we get:(81)$$\Delta V_{k}^{3}(t)=-\int_{0}^{t}{\tilde{d}}_{lump,k}^{T}(t)s_{2k}(t)d\tau -\frac{1}{2\gamma }{\tilde{d}}_{lump,k-1}^{T}(t){\tilde{d}}_{lump,k-1}(t)$$

For the fourth term $\Delta V_{k}^{4}(t)$ in Equation (74)
(82)$$\Delta V_{k}^{4}(t)=\frac{1}{2\varsigma }\int_{0}^{t}\left[tr\left({\tilde{\zeta }}_{k}^{T}(t){\tilde{\zeta }}_{k}(t)\right)-tr\left({\tilde{\zeta }}_{k-1}^{T}(t){\tilde{\zeta }}_{k-1}(t)\right)\right]d\tau$$

From Lemma 1, it can be known that:(83)$$\begin{array}{r} tr\left({\tilde{\zeta }}_{k}^{T}(t){\tilde{\zeta }}_{k}(t)\right)-tr\left({\tilde{\zeta }}_{k-1}^{T}(t){\tilde{\zeta }}_{k-1}(t)\right)=2tr\left[{\left({\hat{\zeta }}_{k-1}(t)-{\hat{\zeta }}_{k}(t)\right)}^{T}\left(\zeta_{d}(t)-{\hat{\zeta }}_{k}(t)\right)\right]\\ -tr\left[{\left({\hat{\zeta }}_{k}(t)-{\hat{\zeta }}_{k-1}(t)\right)}^{T}\left({\hat{\zeta }}_{k}(t)-{\hat{\zeta }}_{k-1}(t)\right)\right] \end{array}$$

Using Equation (83) and the iterative learning law (65), Equation (82) can be simplified as:(84)$$\Delta V_{k}^{4}=-\int_{0}^{t}tr\left(s_{2k}(t)\Phi_{k}^{T}(t){\tilde{\zeta }}_{k}(t)\right)d\tau -\frac{\varsigma }{2}\int_{0}^{t}{\left(\left\|\Phi_{k}(t)\right\|\left\|s_{2k}(t)\right\|\right)}^{2}d\tau$$

Combining Equations (76), (79), (81), and (84), we have:(85)$$\begin{array}{l} \Delta \Omega_{k}(t)\leq -k_{1}\int_{0}^{t}{\left\|s_{1k}(t)\right\|}^{2}d\tau -\int_{0}^{t}k_{2}{\left\|s_{2k}(t)\right\|}^{2}d\tau -\int_{0}^{t}k_{3}\left\|s_{2k}(t)\right\|d\tau \\ \;-\frac{1}{2}\log\frac{\psi^{2}(t)}{\psi^{2}(t)-s_{1k-1}^{T}(t)s_{1k-1}(t)}-\frac{1}{2\gamma }{\tilde{d}}_{lump,k-1}^{T}(t){\tilde{d}}_{lump,k-1}(t)\\ \;-\frac{\varsigma }{2}\int_{0}^{t}{\left(\left\|\Phi_{k}(t)\right\|\left\|s_{2k}(t)\right\|\right)}^{2}d\tau -\frac{1}{2}s_{2k-1}^{T}(t)M_{xk-1}s_{2k-1}(t) \end{array}$$

**Remark** **10.***The equivalent transformation*$s_{2k}^{T}(t){\tilde{\zeta }}_{k}^{T}(t)\Phi_{k}(t)=tr\left(s_{2k}(t)\Phi_{k}^{T}(t){\tilde{\zeta }}_{k}(t)\right)$*is implemented when the above formula is simplified*.

With Equation (85) we can conclude that $\Omega_{k}(t)$ is monotonically nonincreasing in the iterative domain. Therefore, when $\Omega_{0}(t)$ is bounded, the CEF $\Omega_{k}(t)$ is bounded at any number of iterations. Since the CEF conforms to the general properties of the sequence in the iterative domain, it can be obtained from the definition of the sequence.
(86)$$\Omega_{k}(t)=\Omega_{0}(t)+\sum_{i=1}^{k}\Delta \Omega_{k}(t)$$

Taking the derivative of $\Omega_{0}(t)$ with respect to time T. Then, Equation (67) is combined by Equation (73). Finally, the following result can be easily obtained as:(87)$$\begin{array}{l} {\dot{\Omega }}_{0}(t)={\dot{V}}_{0}^{1}(t)+{\dot{V}}_{0}^{2}(t)+{\dot{V}}_{0}^{3}(t)+{\dot{V}}_{0}^{4}(t)\\ \;=-k_{1}{\left\|s_{10}(t)\right\|}^{2}-k_{2}{\left\|s_{20}(t)\right\|}^{2}-k_{3}\left\|s_{20}(t)\right\|\\ \;+tr\left(s_{20}(t)\Phi_{0}^{T}(t){\tilde{\zeta }}_{0}(t)\right)+\frac{1}{2\varsigma }tr\left({\tilde{\zeta }}_{0}^{T}(t){\tilde{\zeta }}_{0}(t)\right) \end{array}$$

The iterative learning law ${\hat{\zeta }}_{0}(t)=\varsigma \Phi_{0}(t)s_{20}^{T}(t)$ is substituted into Equation (87), we get:(88)$${\dot{\Omega }}_{0}(t)=-k_{1}{\left\|s_{10}(t)\right\|}^{2}-k_{2}{\left\|s_{20}(t)\right\|}^{2}-k_{3}\left\|s_{20}(t)\right\|+\frac{1}{\varsigma }tr\left(\zeta_{d}^{T}(t){\tilde{\zeta }}_{0}(t)\right)-\frac{1}{2\varsigma }tr\left({\tilde{\zeta }}_{0}^{T}(t){\tilde{\zeta }}_{0}(t)\right)$$

In the light of dynamic Equation (33) and the iterative learning law (65), the ideal parameter $\zeta_{d}(t)$ is bounded, and the parameter error matrix ${\tilde{\zeta }}_{0}(t)$ of the 0th iteration is continuously bounded within $\left[0,T\right]$. Hence, ${\dot{\Omega }}_{0}(t)$ exists as an upper bound. The following conclusion can be drawn as:(89)$$\Omega_{0}(t)=\Omega_{0}(0)+\int_{0}^{t}{\dot{\Omega }}_{0}(t)d\tau$$

Ultimately, it can be concluded that $\Omega_{k}(t)$ is strictly bounded in the iterative domain.

**Step 2:** Demonstrate that the system state $s_{1k}(t),s_{2k}(t)$ converges. From Equation (86), the following conclusion can be easily reached.
(90)$$\Omega_{k}(t)-\Omega_{0}(t)=\sum_{i=1}^{k}\Delta \Omega_{k}(t)\leq 0$$

Since each term of the CEF $V_{k}^{i}(k),i=1,2,3,4$ is a non-negative definite matrix, we get:(91)$$\begin{array}{l} \sum_{i=1}^{k}\Delta \Omega_{k}(t)\leq -k_{1}\int_{0}^{t}{\left\|s_{1k}(t)\right\|}^{2}d\tau -\int_{0}^{t}k_{2}{\left\|s_{2k}(t)\right\|}^{2}d\tau -\int_{0}^{t}k_{3}\left\|s_{2k}(t)\right\|d\tau \\ \;-\frac{1}{2}\log\frac{\psi^{2}(t)}{\psi^{2}(t)-s_{1k-1}^{T}(t)s_{1k-1}(t)}-\frac{1}{2\gamma }{\tilde{d}}_{lump,k-1}^{T}(t){\tilde{d}}_{lump,k-1}(t)\\ \;-\frac{\varsigma }{2}\int_{0}^{t}{\left(\left\|\Phi_{k}(t)\right\|\left\|s_{2k}(t)\right\|\right)}^{2}d\tau -\frac{1}{2}s_{2k-1}^{T}(t)M_{xk-1}s_{2k-1}(t)\leq 0 \end{array}$$

In conclusion, both $\left\|s_{1k}(t)\right\|$ and $\left\|s_{2k}(t)\right\|$ will eventually be forced to 0 when k→∞. It can be seen from Equation (59) that when $\underset{k\to \infty }{\lim}\left\|s_{1k}(t)\right\|=\underset{k\to \infty }{\lim}\left\|s_{2k}(t)\right\|=0$, the BLF satisfies the condition ${\dot{V}}_{1k}(t)\leq 0$ where $s_{1k}(t)$ is constrained by the time-varying constraint function ψ(t) during the convergence process. Thus far, the convergence proof process of the closed-loop system has been fully demonstrated.

## 5. Simulation Results

To evaluate the validity and superiority of the proposed scheme, the performance of the model and controller was verified via cosimulation. Above all, the cosimulation environment of a virtual prototype was configured by utilizing the ANSYS/ADAMS/MATLAB software. Cosimulation experiments of the virtual prototype were run on a workstation with GTX1050Ti GPU, 16G memory, and an Intel Core i5 CPU for implementation. Then, the expected sinusoidal trajectory with the initial error was employed as the control input of each PMA, and comparative studies with other algorithms were carried out.

In this paper, the finite element analysis of PMA was first carried out by ANSYS software. Subsequently, the generated flexible body was imported into ADAMS software to connect with the rigid body structure. Secondly, the parameters of the multisection continuum arms and the physical simulation environment can be flexibly configured through ADAMS. Finally, the numerical model of the controlled object was generated through the control plants and imported into MATLAB/Simulink for cosimulation.

**Remark** **11.***To further increase the uncertainty and external disturbance of the control system, the bottom of the continuum arms is connected to the SCARA type 2-DOF manipulator. The rigid-flexible coupling system is partially different from the structure shown in*Figure 1*. Nonetheless, the final conclusion is not impacted by model simplification. The model of the cosimulation is executed, as shown in*Figure 9.

**Remark** **12.***To simulate the flexible deformation process of the continuum arms, a single PMA is transferred into ANSYS for preprocessing. First, the material properties of PMA are composed of super-elastic rubber material, which guarantees the flexibility of the pneumatic muscle for one thing, and restrains the radial deformation of the pneumatic muscle during inflation for another. After the material parameters of PMA are obtained, mesh division is performed. Afterwards, the upper and lower ends of the PMA are coupled with mass21 for rigid regions* [32,33]*. Eventually, the preprocessing model (.mnf) is generated and imported into ADAMS (as shown in*
Figure 10*).*

For the multisection continuum arms system, the gravity direction is specified as the negative direction of the *Y*-axis. The material property of the rigid part is defined as structural steel. The command trajectory $x_{d}(t)$ of the end of the multisection continuum arms is set as follows:(92)$$x_{d}(t)=a_{0}+a_{1}*\cos\left(t*w\right)+b_{1}*\sin(t*w)$$
where $\;a_{0}=-0.03537$, $a_{1}=0.04735$, $b_{1}=0.04688$, and w=5.7702 The IBSF is set as $F_{id}=3$, and the man-machine interaction force is selected as $F_{\mathrm{int}}=5\sin\left(1.5\pi ft\right)+4$. Moreover, the impedance coefficient is selected as K=100, B=15, and H=0.22. In the light of Equation (48), the adjustment amount $x_{e}(t)$ of the command trajectory is obtained as follows:(93)$$x_{e}(t)=a*\exp\left(b*t\right)+c*\exp(d*t)$$
where a=−0.00668, b=0.05365, c=0.006695, and d=−7.35 The desired trajectory of the end of the continuum arms is obtained by Equation (49).
(94)$$x_{r}(t)=a_{0}+a_{1}*\cos\left(t*w\right)+b_{1}*\sin(t*w)$$
where $a_{0}=-0.02467$, $a_{1}=0.02465$, $b_{1}=0.07891$, and w=0.1532 The desired trajectory $x_{ck}(t)\in R^{2\times 1}$ of the SCARA manipulator is set as:(95)$$x_{ck1,2}(t)=\sin\left(\frac{\pi t}{50}\right)$$

The desired motion trajectory $x_{ck}(t)$ of each PMA in the joint space is obtained from the kinematic inverse model proposed in Section 2.1.2, and its mathematical expression is:(96)$$\left\{\begin{array}{c} x_{ck3,4,7,8}(t)=0.2150-\sin\left(\frac{\pi t}{25}\right)*0.05\\ x_{ck5,6,9,10}(t)=0.2150-\sin\left(\frac{\pi t}{50}\right)*0.05 \end{array}\right.$$

To test the effectiveness of the trajectory reconstruction method in the presence of initial errors, the initial joint position of the multisection continuum arms and the SCARA manipulator in the kth iteration is set to $x_{1k}(0)=x_{2k}(0)={\left[0,\cdots ,0\right]}^{T}\in R^{10\times 1}$. It can be seen that the initial value of the desired trajectory is not equal to the system state. Afterwards, the reconstructed desired trajectory $x_{gk}(t)\in R^{10\times 1}$ is obtained by Equation (50).

The corresponding control parameters are listed in Table 2.

**Remark** **13.***The selection of control parameters is supposed to be abided by the following approaches. It should be emphasized that the control variate method is adopted for the parameter adjustment. Firstly, the adaptive step*γ*can be appropriately increased, which can effectively compensate for lumped disturbance and smooth the input torque in the iterative domain. Then, increasing the iterative learning step*ς *can speed up the convergence of the iterative domain, generally not exceeding 1. Moreover, increasing*$k_{1}$*slowly is capable of enhancing the system convergence accuracy and efficiency but has a negative impact on system stability. Correspondingly, increasing*$k_{2}$*can stabilize the system and ease chatter phenomena in the time domain. Particularly,*$k_{2}$*is greater than*$k_{1}$*and*$k_{3}$*should not exceed 1. Last but not least, the formulation of the time-varying constraint function counts on the desired maximum error in practical systems*.

To validate the superiority of the proposed control scheme, comparative experiments are conducted by designing an impedance PD controller and an impedance iterative learning sliding mode controller (IILSMC) in this paper. The impedance PD control parameters are specified as follows:(97)$$\left[\begin{array}{c} P\\ D \end{array}\right]=\left[\begin{array}{cccccccccc} 10^{-4} & 10^{-4} & 10^{-3} & 10^{-3} & 10^{-3} & 10^{-3} & 10^{-3} & 10^{-3} & 10^{-3} & 10^{-3}\\ 10^{-2} & 10^{-2} & 10^{-1} & 10^{-1} & 10^{-1} & 10^{-1} & 10^{-1} & 10^{-1} & 10^{-1} & 10^{-1} \end{array}\right]$$

The IILSMC control law and control parameters are set as follows
(98)$$\left\{\begin{array}{c} u_{k}(t)={\hat{\xi }}_{k}^{T}(t)I_{k}(t)+{\hat{d}}_{lump,k}(t)+k_{1}s_{k}(t)+k_{2}\mathrm{sgn}(s)\\ {\dot{\hat{d}}}_{lump,k}(t)=s_{k}(t)\\ {\hat{\xi }}_{k}(t)={\hat{\xi }}_{k-1}(t)+qI_{k}(t)s_{k}^{T}(t)\;,\;{\hat{\xi }}_{-1}(t)=0 \end{array}\right.$$
where $s_{k}(t)=c_{1}e_{1k}(t)+e_{2k}(t)$, $x_{ak}(t)=c_{1}e_{1k}(t)+{\dot{x}}_{gk}(t)$, and $I_{k}(t)={\left[{\dot{x}}_{ak}^{T}(t),x_{ak}^{T}(t),1\right]}^{T}$ to ensure the persuasion, except that the control gain parameter is specified to $c_{1}={\left[2,2,3,3,3,3,3,3,3,3\right]}^{T}$, q=1, $\gamma ={\left[10,2,1,1,1,1,1,1,1,1\right]}^{T}\times 100$, $k_{1}={\left[10,10,8,8,8,8,8,8,8,8\right]}^{T}$, and $k_{2}=2$, other simulation conditions are the same. The simulation results are depicted in Figure 11, Figure 12, Figure 13, Figure 14, Figure 15, Figure 16, Figure 17 and Figure 18.

The desired motion curve of PMA1 before and after trajectory reconstruction is, respectively, shown in the above figure. The desired motion curve before trajectory reconstruction $x_{ck}(t)$ is the blue curve in the figure, while the desired motion curve after trajectory reconstruction $x_{gk}(t)$ is the red curve in the figure. The initial value of the system after the trajectory reconstruction is equal to the initial value of the desired motion trajectory. Meanwhile, the desired trajectory after the trajectory reconstruction at the closed interval $\left[\rho ,T\right]$ is the same as the original desired trajectory.

The tracking situation of the actual motion trajectories and the desired trajectories of each joint after 10 iterations is depicted in the above figure. It can be gathered that the motion trajectory can accurately track the reconstructed desired trajectory with arbitrary precision at k=10. From Figure 12b,c, it can be found that the actual motion trajectory of PMA is forced to change the motion trajectory and quickly track the desired motion trajectory at approximately 1.8 s on account of the designed BLF. Consequently, the position-tracking error is limited by the given time-varying constraint function ψ(t). Regarding time interval t∈[0,ρ), there exists an obvious tracking error in the joint position because the selected lumped disturbance and time-varying constraint function are based on the reconstructed desired trajectory. Nevertheless, the reconstructed desired trajectory $x_{gk}(t)$ can be accurately tracked by the actual motion trajectory $x_{1k}(t)$ of each joint.

To demonstrate more clearly and intuitively that the tracking error of each joint position is uniformly bounded under the given time-varying constraint function, the error convergence curve of each joint-position tracking is depicted in Figure 13. When the tracking error approaches the constraint functions ψ(t) and −ψ(t) (black dotted lines in the figure) at a certain moment, it will be forcibly limited by the controller within the constraint function, and will never exceed the given time-varying constraint function during the convergence process. In conclusion, the position-tracking error of PMA can be guaranteed to be within 0.002 m and 0.004 m.

The desired trajectory and force balance curve for the end of the multisection continuum arms are shown in Figure 14, Figure 15 and Figure 16. Owing to the introduction of impedance control to correct the command trajectory, the desired trajectory obtained is not a standard sinusoid. Hence, the impedance control can ensure that the ISBF is maintained at a given constant value. By adjusting the impedance coefficient, it was found that reducing the value of the impedance coefficient H,B,K resulted in a smoother input torque, but led to an increase in the adjustment amount of the desired trajectory $x_{e}$.

When the number of iterations k=10, the position-tracking error curves of different control algorithms for PMA 1 and 3 are shown in Figure 17. Compared with IILSMC and PD impedance control, the proposed control scheme has the lowest tracking errors, which do not exceed 0.004 m. Furthermore, the actual motion trajectory has the competence to approximate the desired motion trajectory with arbitrary accuracy as the number of iterations increases. Contrasted by the other two algorithms, the position-tracking error obtained by the PD impedance control is divergent in the time-domain, which can be concluded that the traditional PD control cannot effectively handle the lumped disturbance. In other words, the robustness of the system based on PD control is poor. Quite the reverse, the control requirements are always satisfied by IILSMC and the proposed scheme.

Compared with IILSMC, the proposed algorithm possesses higher control accuracy. Specifically, the more superior the control accuracy of the proposed algorithm is, the more the motion speed of the pneumatic muscle increases (e.g., PMA1). For the proposed control scheme, when the tracking error converges to a given time-varying constraint function, the tracking error is forced to stay within the constraint function by controlling the input torque. The above phenomenon can be visualized in Figure 17a. Hence, in contrast with PD control and IILSMC, the proposed control scheme has a better tracking performance for a class of multisection continuum arms with lumped disturbance.

The cosimulation animation of the continuum arms is depicted in Figure 18. Through the cosimulation of ANSYS/ADAMS/MATLAB software, the deformation cloud and force situation of each PMA can be obtained. Apart from that, the motion attitude of the continuum arms can be simulated by the cosimulation animation in real-time. From the overall motion attitude, it can be drawn that the continuum arm model and its control algorithm designed in this paper are generally reasonable.

## 6. Conclusions

In this paper, a kinematic model of multisection continuum arms based on pneumatic muscle drive is established, where a four-parallel structure is adopted for the single pneumatic node of the continuum arms. To establish an accurate kinematic model, the mapping relationship between the deformation parameters of the continuum node and the length of each PMA is obtained based on the assumption of constant curvature utilizing the geometric method. The final kinematic model obtained has an error of up to 0.013%. For the existence of time-varying lumped disturbance of the multisection continuum arms, a dual closed-loop force-position hybrid control strategy is implemented in this paper. To begin with, the ISBF is maintained at 3 N by virtue of position-based impedance control. Meanwhile, to ensure that the tracking error of the multisection continuum arms is bounded, an adaptive iterative learning backstepping control strategy based on the log-type BLF is employed to achieve each PMA to track the desired trajectory with arbitrary accuracy where the tracking error converges stably to 0.004 m and does not exceed a given time-varying constraint function. Compared with IILSMC, the control accuracy of the proposed algorithm is enhanced by 57.14%. Furthermore, the lumped disturbance of the continuum arms can be effectively suppressed by the adaptive algorithm, where the input torque of each PMA is smooth without a chattering phenomenon. Specifically, the condition that the initial error of iterative learning is zero is released by the trajectory reconstruction approach. For PMA-driven multisection continuum robots, the modeling method proposed in this paper is generalized. Last but not least, for systems with complicated models, high degrees of freedom, and periodic motions, the control scheme proposed in this paper is general, where a mathematical model of the controlled object is not required.

Even so, in practical applications, there are some shortcomings in the proposed scheme which are supposed to be further investigated. For extremely complicated models, the computational efficiency tends to decrease as the amount of memory information surges. In the subsequent study, a specific experimental prototype will be executed to further verify the validity of the proposed theory.

## Figures and Tables

**Figure 1 micromachines-13-01532-f001:**
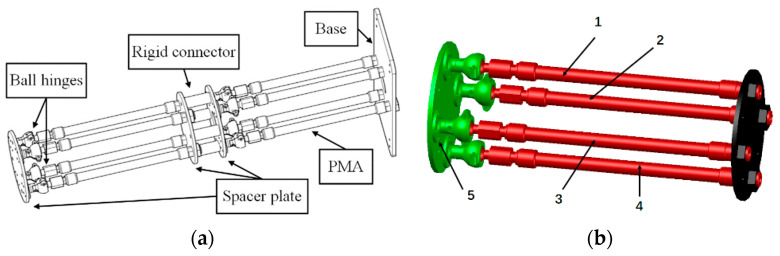
(**a**) Multisection continuum arm structure. (**b**) Single continuum node structure.

**Figure 2 micromachines-13-01532-f002:**
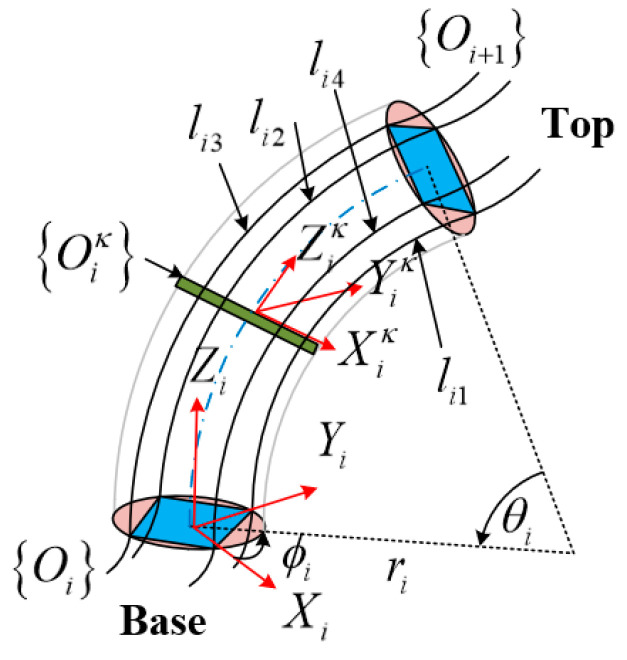
Deformation mode of a single continuum node.

**Figure 3 micromachines-13-01532-f003:**
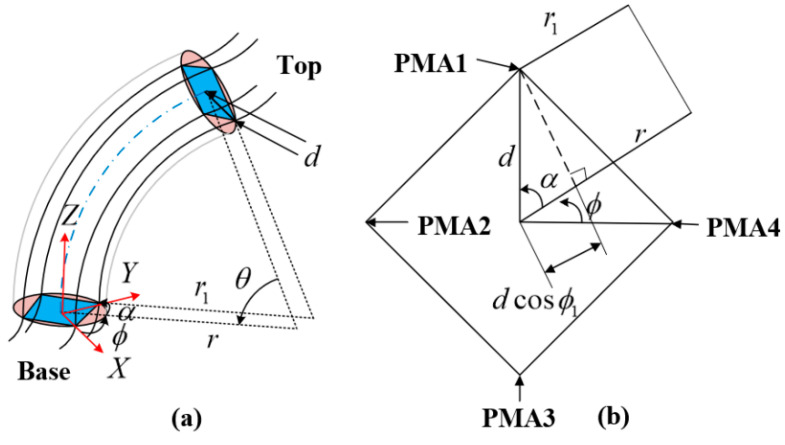
Schematic diagram of the single continuum node: (**a**) the angle of deflection $\phi_{i}$; (**b**) top view.

**Figure 4 micromachines-13-01532-f004:**
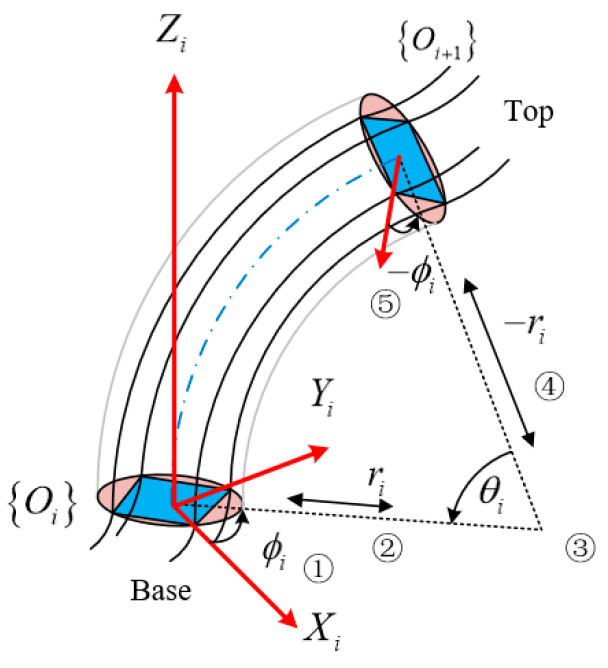
Diagram of single continuum node CPHTM based on enhanced D-H method.

**Figure 5 micromachines-13-01532-f005:**
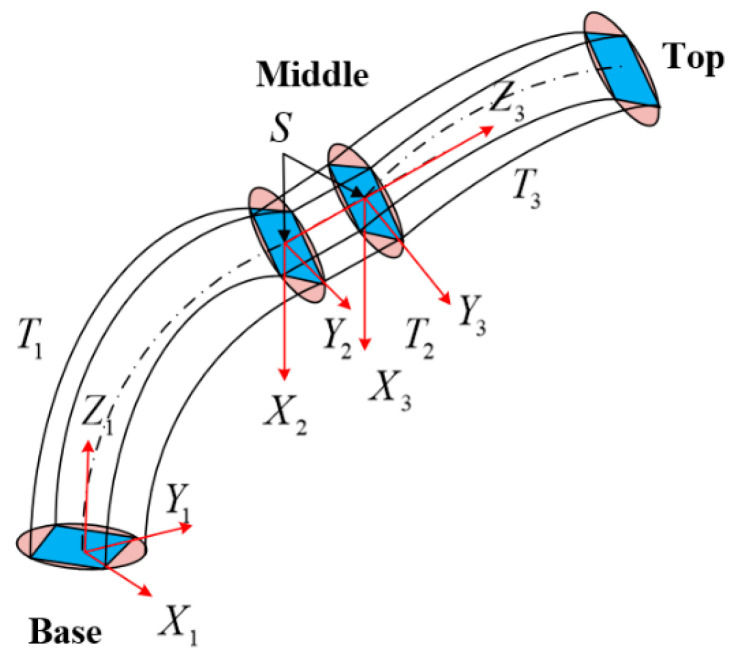
MHTM of multisection continuum arms.

**Figure 6 micromachines-13-01532-f006:**
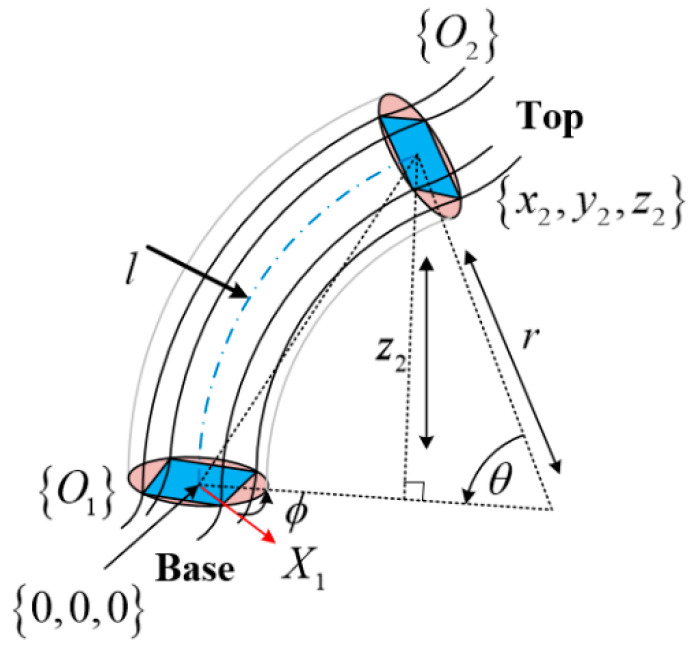
Geometric diagram of the inverse kinematics model of the continuum node.

**Figure 7 micromachines-13-01532-f007:**
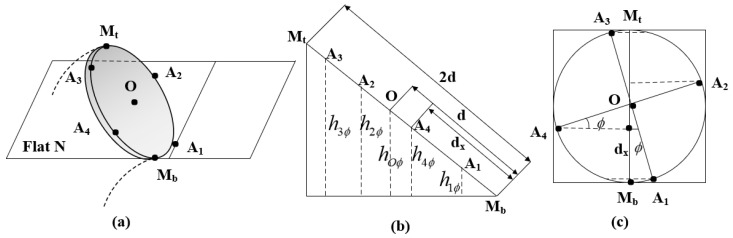
(**a**) Auxiliary plane N. (**b**) Projection view of $h_{1\phi }$,$h_{2\phi }$,$h_{3\phi }$,$h_{4\phi }$,$h_{O\phi }$. (**c**) Projection view of $A_{1}$,$A_{2}$,$A_{3}$,$A_{4}$.

**Figure 8 micromachines-13-01532-f008:**
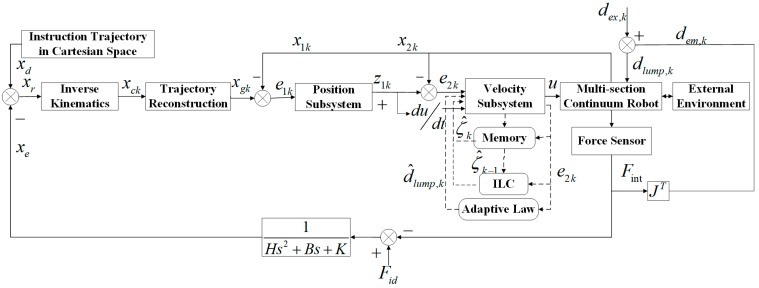
The control flow diagram of the proposed algorithm.

**Figure 9 micromachines-13-01532-f009:**
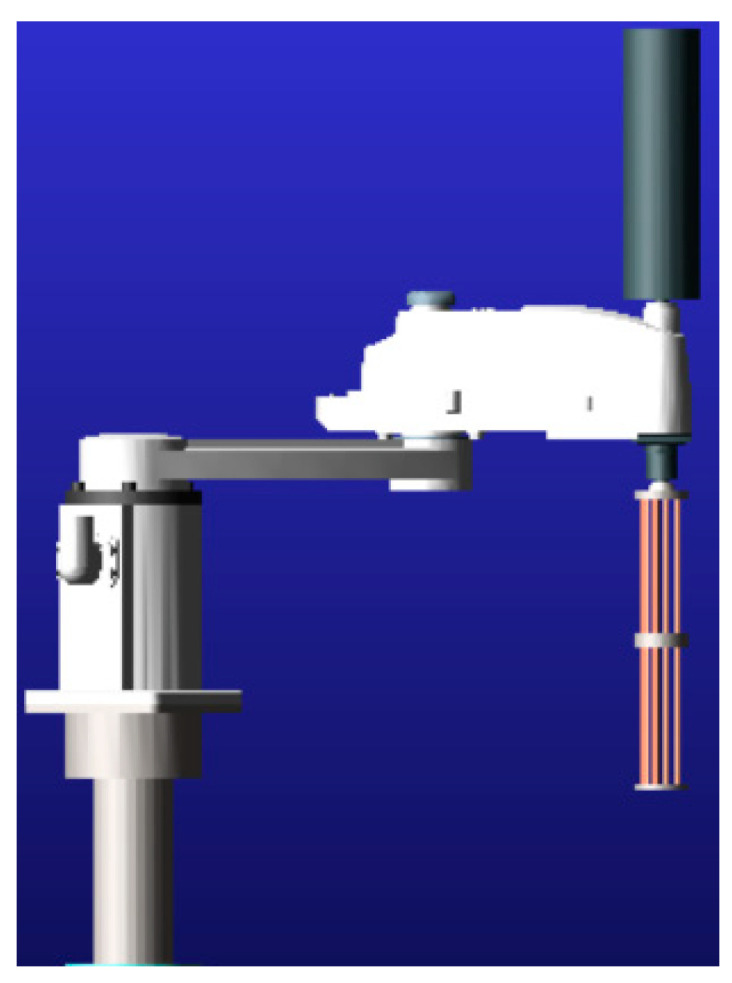
Cosimulation model of continuum robot system.

**Figure 10 micromachines-13-01532-f010:**
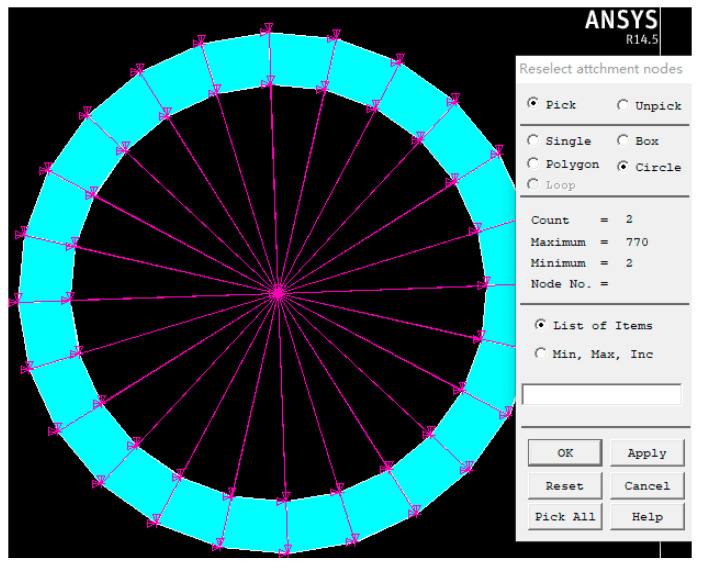
The preprocessing model (.mnf) obtained from ANSYS.

**Figure 11 micromachines-13-01532-f011:**
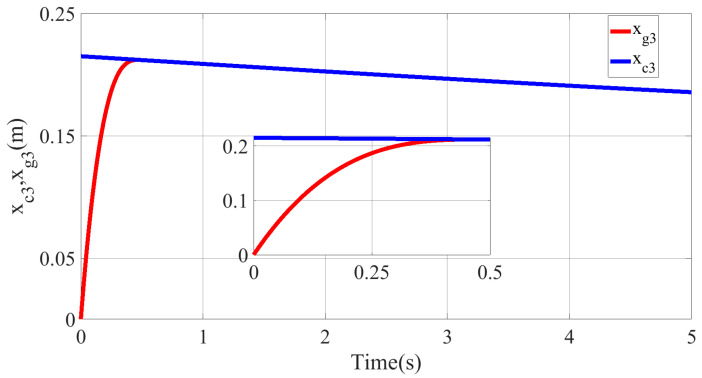
Motion curves before and after trajectory reconstruction.

**Figure 12 micromachines-13-01532-f012:**
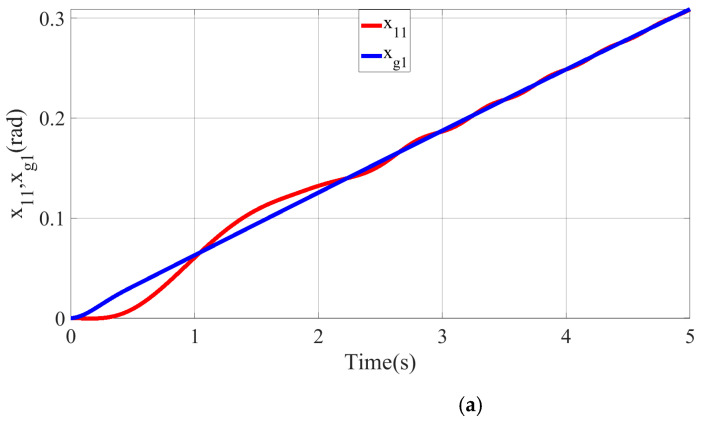
Iterative convergence curve of each joint-position tracking. (**a**) The joint-angle tracking curve of the SCARA manipulator. (**b**) Position-tracking curve of PMA1. (**c**) Position-tracking curve of PMA3.

**Figure 13 micromachines-13-01532-f013:**
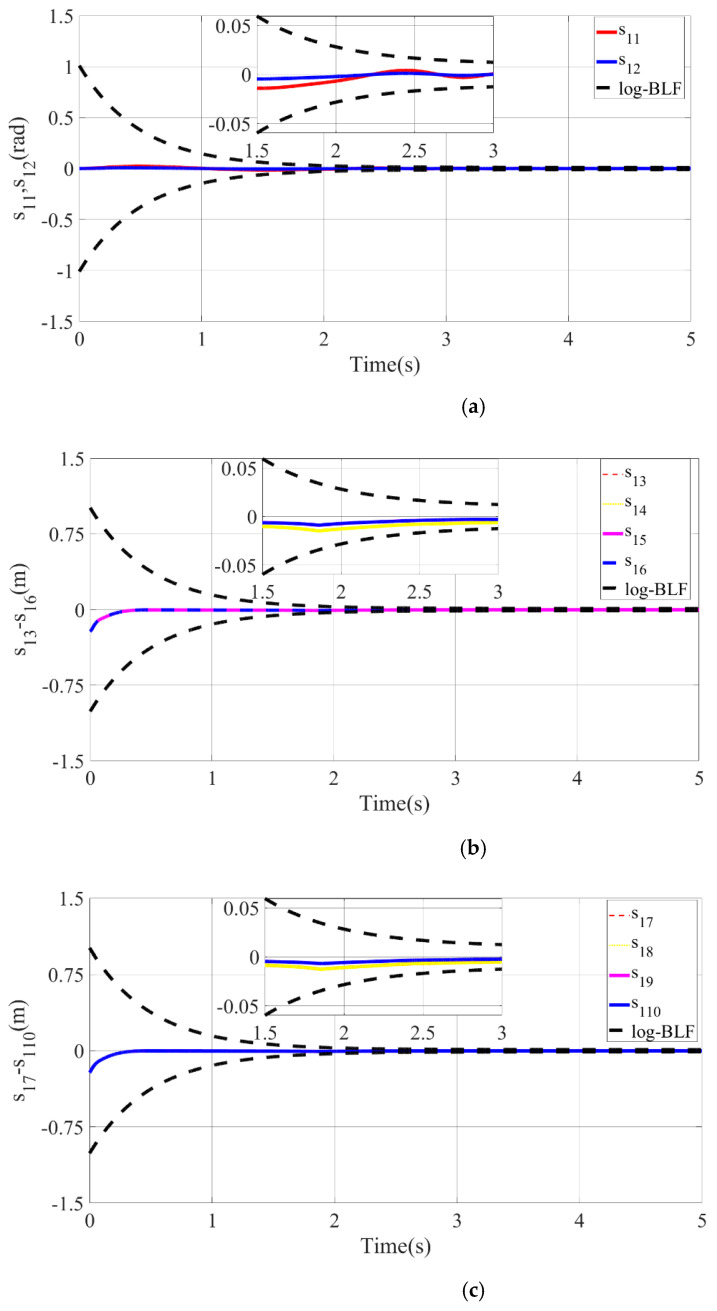
Iterative convergence curve of each joint-position tracking. (**a**) Angle-tracking error of SCARA manipulator in the joint space. (**b**) Position-tracking error of PMA1-4. (**c**) Position-tracking error of PMA5-8.

**Figure 14 micromachines-13-01532-f014:**
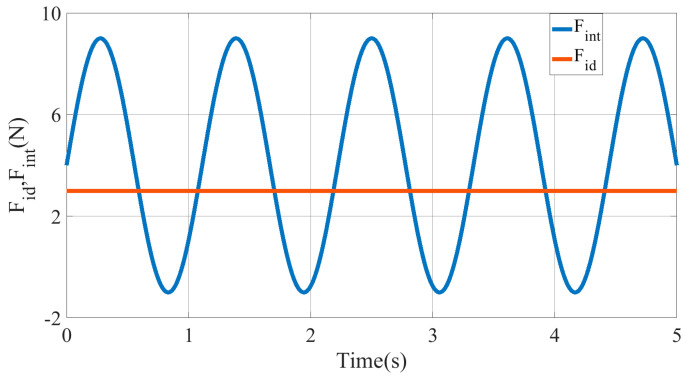
The curve of ISBF $F_{id}$ and human-machine interaction force $F_{\mathrm{int}}$.

**Figure 15 micromachines-13-01532-f015:**
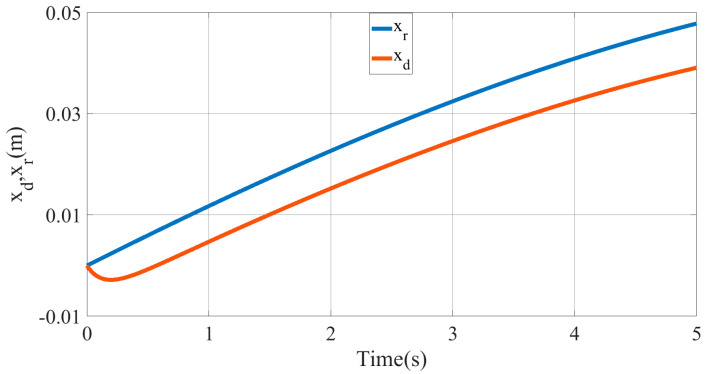
The commanded trajectory $x_{d}$ and the desired trajectory $x_{r}$ curve of the multisection continuum arms in the Cartesian space.

**Figure 16 micromachines-13-01532-f016:**
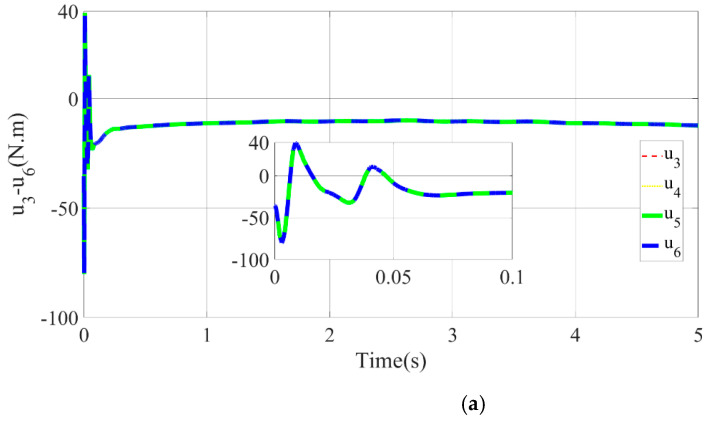
Input torque curve of each PMA. (**a**) Input torque of PMA1-4. (**b**) Input torque of PMA5-8.

**Figure 17 micromachines-13-01532-f017:**
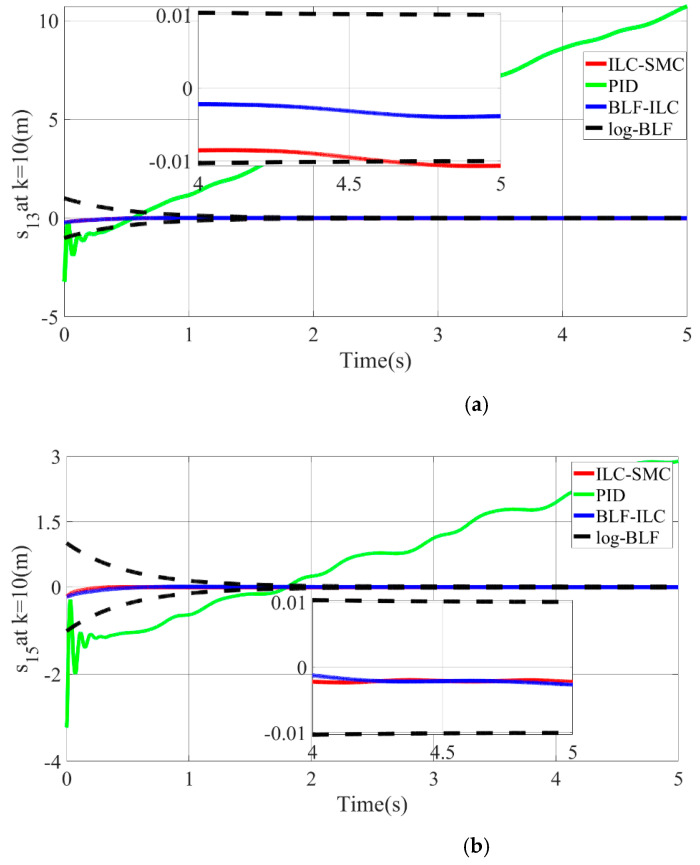
Comparison of simulation results with other control algorithms. (**a**) Position-tracking error curves of different control algorithms for PMA1. (**b**) Position-tracking error curves of different control algorithms for PMA3.

**Figure 18 micromachines-13-01532-f018:**
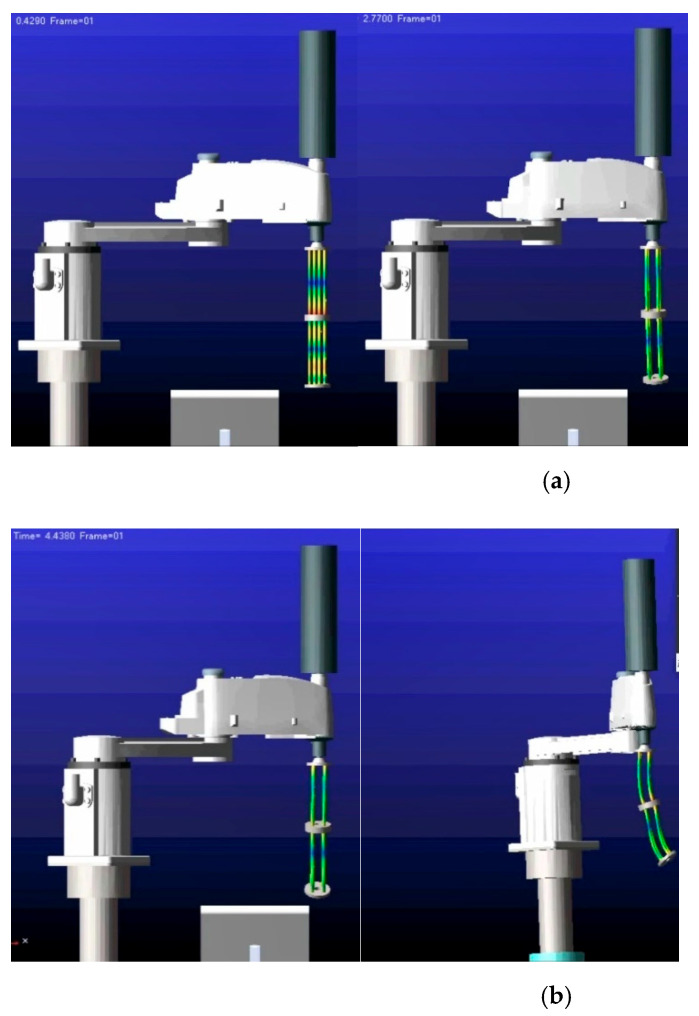
Cosimulation demonstration. (**a**) Cosimulation animation in T=0.42s and T=2.76s; (**b**) cosimulation animation in T=4.44s and T=5s.

**Table 1 micromachines-13-01532-t001:** Kinematic modeling error.

The Actual Length of PMA/m	The Ideal Length of PMA/m	Error/m
0.31254066	0.3125	0.00004066
0.32196066	0.3225	−0.00053934
0.33228767	0.3325	−0.00021233
0.34226136	0.3425	−0.00023864
0.35197250	0.3525	−0.0005275
0.36253373	0.3625	0.00003373
0.31254066	0.3125	0.00004066
0.32196066	0.3225	−0.00053934
0.33228767	0.3325	−0.00021233

**Table 2 micromachines-13-01532-t002:** Control parameters.

Controllers	Parameters
End time of trajectory reconstruction	ρ=0.5
Simulation time	T=5
Iterative learning step	ς=0.5
Adaptive step	γ=0.2
Control gains	$k_{1}=10$, $k_{2}=20$, $k_{3}=0.5$
Number of iterative learning	k=10
Time-varying constraint function	$\psi (t)=e^{-2t}+0.01$
Lumped disturbance	$d_{lump}(t)=0.1*\left(t+\sin(0.1*t)\right)+2$
Auxiliary parameter	$Q_{1}=2$

## Data Availability

All data generated or analyzed during the study are included in the article, and the data that support the findings of this study are openly available.

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
