# Peer review of "Impedance Iterative Learning Backstepping Control for Output-Constrained Multisection Continuum Arms Based on PMA"

_micromachines, 2022, doi:10.3390/mi13091532_

Round 1

Reviewer 1 Report

Overall, the paper is well written, the content organization is logical, and the results are potentially useful. However, there are some problems that should be addressed. The following comments and suggestions are provided for the Authors to further improve their work:

1/ The math notations are weirdly formatted; there seems to be a strange shift up of all formulas, etc.

2/ The form of the abstract should be modified and re-written.

3/ In the introduction part, the literature survey is quite good. However, I think that the authors could enrich the reference section by discussing some new works related to sliding mode control and adaptive robust control methods, especially the dynamic sliding mode control methods, robust sliding mode method, multiple sliding mode methods, and so on, should be included. To help the authors in this direction, I suggest the following reference: design of a non-singular adaptive integral-type finite time tracking control for nonlinear systems with external disturbances, finite-time convergence of perturbed nonlinear systems using adaptive barrier-function nonsingular sliding mode control with experimental validation, adaptive nonsingular terminal sliding mode control for performance improvement of perturbed nonlinear systems, and the introduction should be added to do a better job of explaining the existing methods and why they are or are not valuable. What research gap did you find from previous researchers in your field (it is still partially described, but needs to be expanded and made clearer)?  Mention it in the Novelties section. It will improve the strength of the article.

4/ The motivation and background of wide practical use of the theoretic results presented should be clearly emphasized to facilitate the readers.

5/ If the Lyapunov functions are chosen via the viewpoint of practical application, the authors should give some effective suggestions. What new modifications are introduced in the Lyapunov stability method?

6/ In simulations, lot of parameters are set to certain values, please add more details of how the parameters of the controller are obtained. How the full-state constraints are chosen? It is better to explain how the values of the control parameters in the proposed method are adjusted? Whether these parameters are optimal for simulation results? More explanation and evidence should be given in detail.

7/ Detailed implementation information should be provided (hardware, software, configuration, settings). A detailed discussion of hardware and software applied to the vehicle should be mentioned. Provide specifications of the hardware and software used for simulation of the approach. Because there is not enough data on this paper, the research results on the core idea of this paper seem unreliable.

8/ The presentation of the paper can be improved and the quality of some figures should be enhanced. My disappointment regards figures, to look at them I have to zoom in a lot exploiting their high definition. In my opinion, in a printed version it should be impossible to correctly read some of the figures. So please enlarge them, such as: Fig. 8, Fig. 10, and Fig. 18. Please check the entire figures in the paper.

9/ The analysis in this paper should be supported by experimental results. The authors should use practical systems to validate the proposed methods with experiment results. The validity of these relevant to applications is impossible to judge without experimental testing.

10/ The manuscript writing can be further polished with professional English. The manuscript can be thoroughly revised for grammar check.

Reviewer 2 Report

The submitted research work proposes a complete set of kinematic modeling for a class of PMA-based multisection continuum robots by implementing: i) an impedance iterative learning backstepping control strategy with unknown parameters of the dynamics model and ii) a BLF to ensure that the tracking error is bounded throughout the iterative process considering the time-varying constraint on the tracking error. The Introduction includes a comprehensive literature review, describes sufficiently the problem and reports the limitations/disadvantages of the methods/models that are already available in the literature. The work’s objectives are clearly described and the authors’ original research contribution is highlighted. In general, the manuscript is well-organized. The system modeling is presented in details in Section 2. It can be considered that the length of Section 2 may be too large. However, it contains valuable information that enables the reader to follow the content. Also, it is rich in engineering judgment and authors’ insightful comments. By considering Section 3 and Section 4, it can be said that a holistic approach has been adopted for the controller’s design. The reviewer appreciates that the fact that few practical issues/limitations has been considered during the adopted design approach. The results, presented in Section 5, seem to be convincing and validate the effectiveness of the introduced methodology. Generally-speaking, it can be said the submitted research work would be valuable for engineers that work in the specific area and it could provide reference for future research. There are no major revisions that could be made by considering also the large length of the manuscript. Few minor revisions are the following:

a)      Although the english language and style are fine, moderate english changes are required.

b)     Please re-write the Abstract. The Abstract should be a total of about 200 words maximum. It has to follow the style of structures abstracts, but headings (e.g. background, methods, etc.) should be included.

Round 2

Reviewer 1 Report

Most of the recommendations I made were addressed by the authors, therefore I believe that the paper is now ready for publication.